# *Legionella* metaeffector MavL reverses ubiquitin ADP-ribosylation via a conserved arginine-specific macrodomain

Zhengrui Zhang[1], Jiaqi Fu [2], Johannes Gregor Matthias Rack [3,5], Chuang Li[2], Jim Voorneveld[4], Dmitri V. Filippov [4], Ivan Ahel [3], Zhao-Qing Luo [2] & Chittaranjan Das [1] ✉

ADP-ribosylation is a reversible post-translational modification involved in various cellular activities. Removal of ADP-ribosylation requires (ADP-ribosyl) hydrolases, with macrodomain enzymes being a major family in this category. The pathogen *Legionella pneumophila* mediates atypical ubiquitination of host targets using the SidE effector family in a process that involves ubiquitin ADP-ribosylation on arginine 42 as an obligatory step. Here, we show that the *Legionella* macrodomain effector MavL regulates this pathway by reversing the arginine ADP-ribosylation, likely to minimize potential detrimental effects caused by the modified ubiquitin. We determine the crystal structure of ADP-ribose-bound MavL, providing structural insights into recognition of the ADP-ribosyl group and catalytic mechanism of its removal. Further analyses reveal DUF4804 as a class of MavL-like macrodomain enzymes whose representative members show unique selectivity for mono-ADP-ribosylated arginine residue in synthetic substrates. We find such enzymes are also present in eukaryotes, as exemplified by two previously uncharacterized (ADP-ribosyl)hydrolases in *Drosophila melanogaster*. Crystal structures of several proteins in this class provide insights into arginine specificity and a shared mode of ADP-ribose interaction distinct from previously characterized macrodomains. Collectively, our study reveals a new regulatory layer of SidE-catalyzed ubiquitination and expands the current understanding of macrodomain enzymes.

The Gram-negative pathogen *Legionella pneumophila* is the causative agent for Legionnaires' disease, a severe type of pneumonia resulting from infection of alveolar macrophages. Upon entry into its host cells, the bacterium utilizes its Dot/Icm type-IV secretion system (T4SS) to translocate at least 330 effectors into the host cytosol for its survival and replication[1–3]. Among the variety of strategies employed by these effectors, ADP-ribosylation, in which ADP-ribose (ADPR) from nicotinamide adenine dinucleotide (NAD[+]) is added onto target proteins, has

recently emerged as a means for host subjugation. Previous studies have described three examples of mono-ADP-ribosylation (MARylation) events catalyzed by *Legionella* effectors[4–6]. The SidE effector family, represented by its prototypical member SdeA, targets Rab GTPases associated with the endoplasmic reticulum (for example, Rab33b) and reticulon 4 through a two-step noncanonical ubiquitination pathway involving ADP-ribosylation. In this reaction, ubiquitin (Ub) is first MARylated at Arg42 by the SdeA mono-ADP-ribosyl

[1]Department of Chemistry, Purdue University, West Lafayette, IN 47907, USA. [2]Department of Biological Sciences, Purdue Institute for Inflammation, Immunology and Infectious Disease, Purdue University, West Lafayette, IN 47907, USA. [3]Sir William Dunn School of Pathology, University of Oxford, South Parks Road, OX1 3RE Oxford, UK. [4]Bio-Organic Synthesis, Leiden Institute of Chemistry, Leiden University, 2300 RA Leiden, The Netherlands. [5]Present address: MRC Centre for Medical Mycology, University of Exeter, Geoffrey Pope Building, Stocker Road, EX4 4QD Exeter, UK. ✉e-mail: cdas@purdue.edu

transferase (mART) domain, followed by processing of the MARylated Ub intermediate (ADPR-Ub) by the phosphodiesterase (PDE) domain, also embedded in the same protein in a different part, to modify serine residues of the host targets via phosphoribosyl (PR)-linked Ub[6–8]. The effector Lart1 has been found to MARylate a conserved arginine in a class of glutamate dehydrogenases present in the natural fungal and protist hosts of *Legionella*, resulting in the inhibition of the metabolic enzyme[5]. Lastly, the recently reported Ceg3 effector was found to suppress host mitochondrial ADP/ATP exchange by MARylation of the ADP/ATP translocase (ANT) family[4,9].

Metaeffectors are the effectors used to counteract or regulate the function of other effectors either through direct effector-effector interaction or opposing action on host targets. In *L. pneumophila*, metaeffectors are widespread among the effector arsenal to ensure a balanced control of host cellular processes[10–12]. For example, the phosphoribosyl ubiquitination pathway mediated by the SidE family is regulated by four metaeffectors. The effectors DupA and DupB, using SidE-like PDE domains, regenerate host targets by hydrolytic removal of the phosphoribosylated Ub (PR-Ub) from PR-ubiquitinated proteins[13,14], whereas the pseudokinase effectors, SidJ and its paralog SdjA, directly inactivate the mART domain of SidE enzymes by poly-glutamylation of the catalytic glutamate present within the signature R-S-E motif, thereby preventing ubiquitination by shutting off the initial MARylation step[15–20].

Recently, a *Legionella* macrodomain-containing enzyme, Larg1, was found to be a metaeffector counteracting Ceg3 by reversing the arginine MARylation of the mitochondrial ANT targets[9,21]. Macrodomains are structural modules that serve as ADPR readers or erasers, with three classes primarily associated with catalytic activity, MacroD-type, PARG-like, and ALC1-like macrodomains, functioning mainly as hydrolases of *O*-glycosidic linkages in ADP-ribosylated proteins[22]. Even though TARG1 and DarG in the ALC1-like class were found to cleave the *N*-glycosidic bond from MARylated thymidine on single-stranded DNA substrates[23,24], there is no evidence that these three macrodomain classes can process the ADPR group from arginine *N*-glycosidic linkages on protein substrates. Interestingly, the recent discovery of Larg1 catalyzing reversal of arginine MARylation[9,21] suggests that macrodomain enzymes could also function on this type of MARylation. However, despite a clear sequence difference between Larg1 with known macrodomains[9,21], no homologs of this bacterial effector have been reported, posing the question of whether more macrodomains can function uniquely on arginine MARylation.

In this study, we perform an activity-based proteome-wide search for Ub interactors among *L. pneumophila* effectors and find a functionally uncharacterized effector, MavL (lpg2526), exhibiting Ub-binding property. Further biochemical investigation of this effector reveals that it is a macrodomain enzyme capable of reversing MARylation on Arg42 of Ub introduced by the SidE effector family. *In cellulo*, MavL behaves as a metaeffector regulating the SidE-mediated noncanonical ubiquitination by recovering Ub. Our structural studies reveal distinct ADP-ribose binding features of MavL compared to the known macrodomains. In addition, bioinformatical analysis suggests a class of MavL-related macrodomain enzymes, DUF4804, with specificity towards arginine MARylation, including the recently reported *Legionella* effector Larg1 as a distant homolog of this class. Overall, this study uncovers a new regulatory layer of SidE-mediated noncanonical ubiquitination and enhances the current scope of macrodomain enzymes.

## Results

### MavL removes ADPR from ADPR-Ub generated by SidE effector family

In response to the diversity of roles played by the Ub system in mediating innate immune response, xenophagy, and cellular trafficking, pathogens have evolved a variety of means to confront and even manipulate the Ub system of their eukaryotic hosts[25]. Previous studies have shown an extensive interaction network between the host Ub system and *Legionella* effectors, including the SidE effector family, MavC, several eukaryote-like Ub E3 ligases, and deubiquitinases[26], raising the possibility of more undiscovered Ub-related effectors in the rather expansive *L. pneumophila* arsenal. To investigate this, we performed a proteome-wide search of Ub interactors in *L. pneumophila* (Fig. 1A) using Ub-derived activity-based probes (Ub-ABPs): HA-Ub-propargylamine[27] (HA-Ub-Prg), HA-Ub-vinylmethylester[28] (HA-UbVME), and HA-Ub-vinylsulfone[28] (HA-Ub-VS). In the same experiment, we included buffer and HA-Ub as two separate controls. Following data processing, we manually inspected and selected the effectors that were enriched by at least one of the probes compared to the controls (Supplementary Data 3). Using this approach, we successfully captured a handful of known Ub-related *Legionella* effectors, including the SidE family ligases, deubiquitinase LotA, transglutaminase MavC, and E3 ligases SidC and SdcA. In the same dataset, we found MavL (lpg2526) as an effector that was robustly captured by HA-UbVME and HA-Ub-VS (Supplementary Data 3). As full-length recombinant MavL behaved poorly in expression and purification, we chose to proceed with MavL$_{42-435}$, a construct that was recently crystallized and its structure reported[29]. Consistent with our proteomics results, the recombinantly purified MavL$_{42-435}$ formed Ub adducts with UbVME and Ub-VS (Fig. 1B). In general, covalent modifications of a protein by Ub-ABPs require both Ub binding and a reactive cysteine proximal to C-terminus of the bound Ub probe. Mutation of the only cysteine (C226) in MavL$_{42-435}$ to alanine abolished UbVME and Ub-VS modifications on MavL (Fig. 1B), showing that both probes indeed react with the cysteine residue of MavL. We further confirmed the direct interaction between MavL$_{42-435}$ and Ub using isothermal titration calorimetry (ITC) which revealed a $K_d$ of 88.3 μM (Fig. 1C). Together, our data suggest that MavL features a Ub-binding site with an affinity typical of commonly observed interaction between Ub and its binding partners, of either eukaryotic or prokaryotic origin[30].

To explore the function of MavL, we used the recently published apo MavL structure[29] (PDB: 6OMI) and performed a protein-fold similarity search using DALI[31], which suggested that the structure of MavL is similar to that of a recently reported *Legionella* macrodomain-type (ADP-ribosyl)hydrolase Larg1[9,21] (PDB: 7W3S), with a Z-score of 24.8. Two other macrodomain proteins, the PARG-like poly-(ADP-ribosyl) glycohydrolase from *Thermomonospora curvata* (*Tc*PARG, PDB: 3SIH) and the MacroD-type MacroD1 from *Homo sapiens* (PDB: 6LH4), were also picked up in the search, with Z-scores of 5.8 and 4.4, respectively. The presence of macrodomain in MavL indicates that it may serve as an ADPR binder or even harbor (ADP-ribosyl)hydrolase activity. Indeed, ITC experiments suggested that MavL binds to ADPR with a $K_d$ of 25.8 μM (Fig. 1D), consistent with the reported $K_d$ of 13 μM[29]. Thus, MavL features both Ub and ADPR binding, presumably as elements of substrate recognition of a putative ADPR processing enzyme. A tighter interaction with ADPR compared to Ub suggests that MavL recognition of ADPR-Ub is dominated by the nucleotide part.

Given the fact that SidE effector family in *L. pneumophila* MARylates Ub on Arg42, we hypothesized that MavL may function to regulate SidE function through removal of ADPR from ADPR-Ub. To test this hypothesis, we generated ADPR-Ub and the related derivative PR-Ub using SdeA$^{mART}$ and SdeA$^{PDE+mART}$ [7,8] and incubated them with MavL. These reactions were analyzed by native PAGE and ESI-MS (Fig. 1E and Supplementary Fig. 1), which showed that MavL can process ADPR-Ub, but not PR-Ub, to recover native Ub. Previous studies have shown two other *Legionella* effectors, DupA and DupB, can generate PR-Ub from ADPR-Ub and PR-ubiquitinated proteins[13,14]. We compared the activity of MavL to DupA and DupB with respect to ADPR-Ub processing using native PAGE and SDS-PAGE (Coomassie Blue stain and phosphoprotein stain) (Fig. 1F). While PR-Ub formation from ADPR-Ub catalyzed by DupA and DupB could be readily detected, only native Ub was

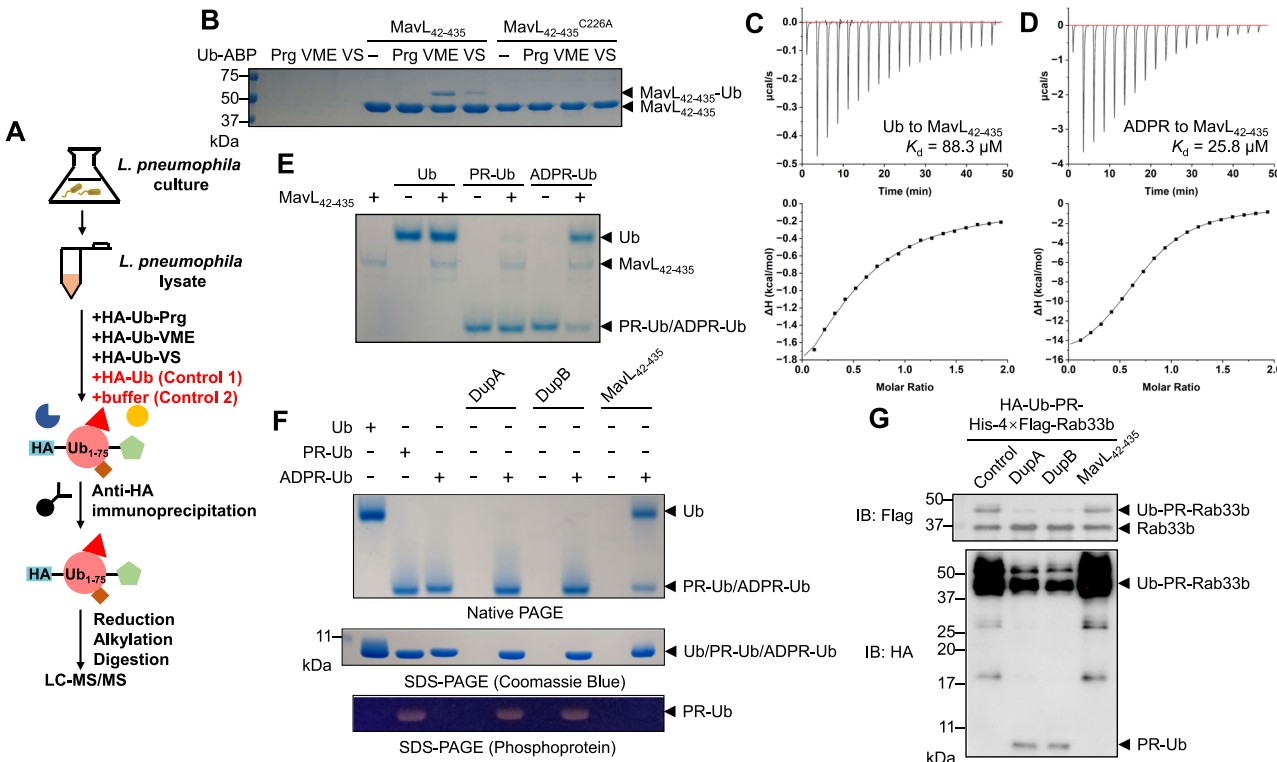

**Fig. 1 | The Ub-interacting *Legionella* effector MavL removes ADPR from ADPR-Ub. A** Workflow of proteome-wide identification of Ub interactors in *Legionella pneumophila*. Red triangle, brown square, and green pentagon represent *L. pneumophila* proteins captured by Ub-ABPs. **B** SDS-PAGE showing MavL-Ub adduct formation upon modifications by UbVME and Ub-VS. Mutation of C226 to alanine in MavL abolished these modifications. Experiments were performed three times independently with similar results. **C, D** Isothermal titration calorimetry profiles of **C** Ub to MavL$_{42-435}$ and **D** ADPR to MavL$_{42-435}$. Raw data were integrated and fitted using a one-binding site model to determine the $K_d$ values. **E** Native PAGE showing the regeneration of Ub from ADPR-Ub, not PR-Ub, by MavL. Reactions were visualized by Coomassie Blue staining. Controls of Ub variants and enzymes alone were included. Experiments were performed three times independently with similar results. **F** Comparison of ADPR-Ub processing by DupA, DupB, and MavL. Reactions were analyzed by native PAGE and SDS-PAGE and visualized by Coomassie Blue staining. PR-Ub was analyzed by SDS-PAGE with phosphoprotein staining. Controls of Ub variants and enzymes alone were included. Experiments were performed three times independently with similar results. **G** Comparison of Ub-PR-Rab33b processing by DupA, DupB, and MavL. Rab33b (His-4 × Flag-tagged) was PR-ubiquitinated with HA-tagged Ub by SdeA and treated with DupA, DupB, or MavL. Reactions were analyzed by anti-Flag and anti-HA immunoblotting to show deubiquitination of Rab33b. Experiments were performed three times independently with similar results.

produced upon MavL-catalyzed hydrolysis of ADPR-Ub. To further probe if MavL can act on PR-ubiquitinated Rab33b (Ub-PR-Rab33b), we generated Ub-PR-Rab33b with HA-Ub and His-4×Flag-tagged Rab33b using SdeA. Upon incubation of Ub-PR-Rab33b with either DupA, DupB, or MavL, we found that MavL cannot hydrolyze PR-Ub from Ub-PR-Rab33b, further suggesting a distinct enzymatic activity compared to DupA and DupB (Fig. 1G). Since MavL is unable to process PR-Ub or Ub-PR-Rab33b, it most likely cleaves off the ADPR group in one single reaction rather than through two consecutive reactions involving PR-Ub as an intermediate. Together, our data suggest that MavL is a macrodomain-type (ADP-ribosyl)hydrolase hydrolyzing the *N*-glycosidic bond on MARylated Arg42 of Ub.

## Metaeffector properties of MavL

To better understand the role of MavL in *L. pneumophila* infection, we first tested if MavL is required for the optimal virulence of *L. pneumophila* by infecting Raw264.7 macrophages separately with WT *L. pneumophila*, a *dotA*⁻ mutant (defective in translocating effectors), or the Δ*mavL* mutant. Like many *L. pneumophila* effectors, deletion of MavL did not cause an appreciable defect in bacterial intracellular replication (Fig. 2A), indicating this effector is dispensable for bacterial replication in our infection model. Nevertheless, the fact that MavL hydrolyzes the ADPR-Ub produced by SdeA suggests that it may function as a previously undiscovered metaeffector against the SidE family. To probe this, we first infected HEK293T cells expressing HA-Ub (through transient transfection) with WT or Δ*mavL L. pneumophila*. We

found that deletion of MavL in *L. pneumophila* results in an appreciably higher level of ADPR-Ub accumulation in infected mammalian cells (Fig. 2B, C) compared to infection with the WT strain. Expression of MavL, but not the catalytically impaired D333A mutant (see below), in Δ*mavL L. pneumophila* (Fig. 2B, C), causes an attenuation of the ADPR-Ub level, providing evidence in support of enzymatic function of MavL on ADPR-Ub under *L. pneumophila* infection condition. We further tested if MavL consequently decreases PR-ubiquitination on Rab33b by infecting HEK293T cells expressing Flag-Rab33b with *L. pneumophila*. Here, we found that co-transfection with a plasmid encoding MavL, but not the D333A mutant, caused suppression of the PR-ubiquitination of Rab33b (Fig. 2D). Furthermore, we tested if MavL reduced overall PR-ubiquitination level in HEK293T cells by expressing HA-tagged Ub^AA (the last two Gly changed to Ala to eliminate the complication of canonical ubiquitination). Indeed, we observed a lower PR-ubiquitination level in the cellular lysate when MavL, but not the D333A mutant, was co-expressed with HA-Ub^AA (Fig. 2E). Collectively, these results indicate that MavL antagonizes the SidE family, which is further validated by our observations that MavL, not the D333A mutant, can rescue the toxicity in yeast caused by WT SdeA or the PDE-inactive SdeA^H277A mutant (both constructs capable of producing ADPR-Ub) (Fig. 2F). Overall, our results here suggest that MavL functions as a metaeffector against the SidE family by counteracting the production of ADPR-Ub (Fig. 2G).

Before this study on MavL, Voth and colleagues[29] showed that MavL interacts with a host E2 Ub-conjugating enzyme, UBE2Q1. As this

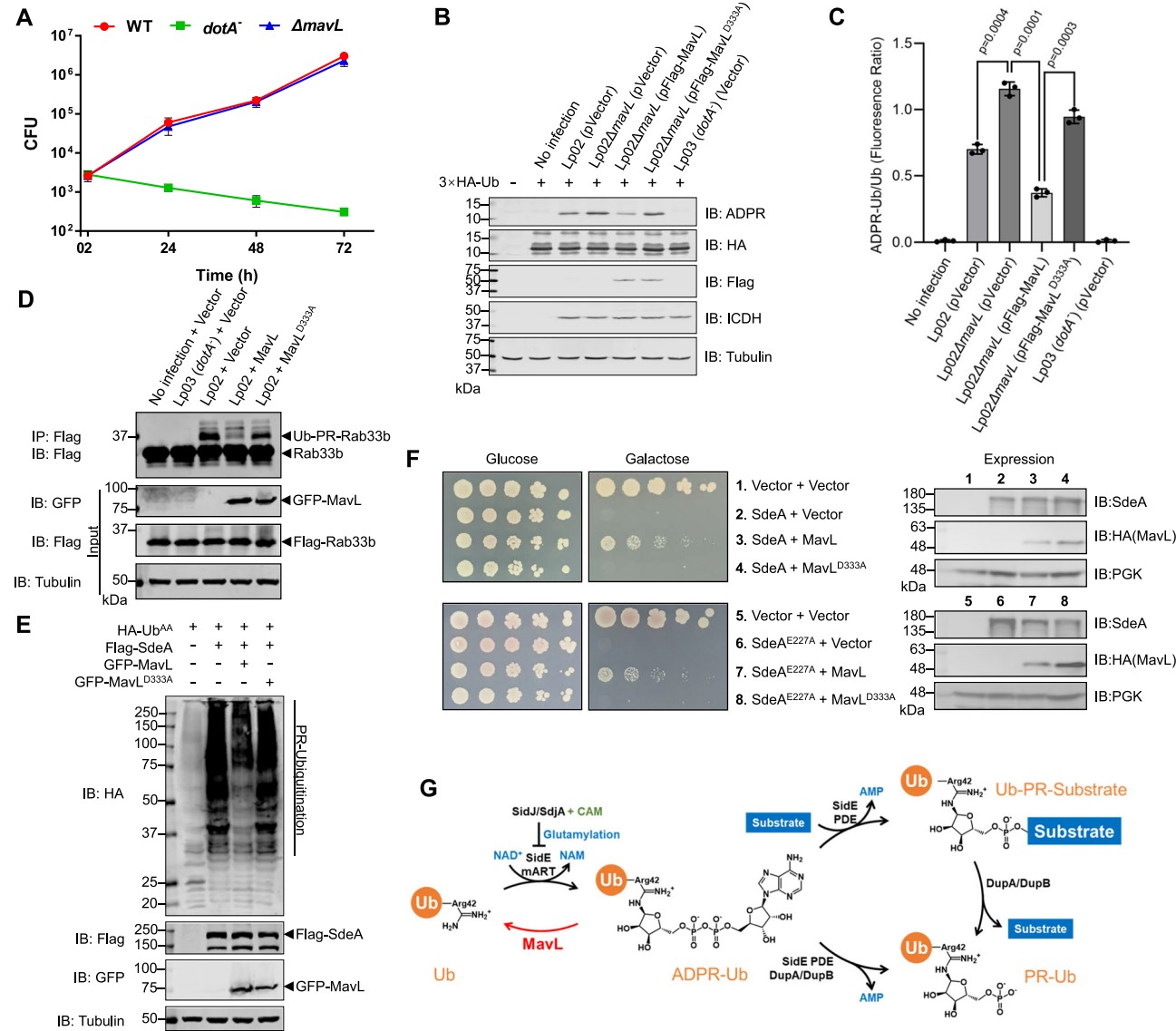

**Fig. 2 | MavL is a metaeffector counteracting the SidE family. A** Deletion of *mavL* did not affect intracellular growth of *L. pneumophila*. Raw264.7 macrophages were infected with the indicated bacterial strains, and the colony-forming units (CFU) of each strain were monitored at 24-h intervals. *n* = 3 biologically independent samples. Error bars: standard deviation (SD) of the mean. **B**, **C** MavL reduces ADP-ribosylation on Ub under infection condition. HEK293T cells transfected with 3 × HA-Ub and antibody receptor FCγRII were infected with different *L. pneumophila* strains. At 2-h post infection, the HEK293T cells were lysed and ADP-ribosylation level on Ub was analyzed by immunoblotting against ADPR (shown in **B**). Expression of HA-Ub and translocation of Flag-tagged MavL were shown by immunoblotting against HA-tag and Flag-tag, respectively. Quantitation of three independent experiments was shown in (**C**). ADPR-Ub to Ub ratio was calculated by dividing the fluorescence intensity of anti-ADPR immunoblotting by that of anti-HA immunoblotting. Error bars: standard error of the mean (SEM). Statistical analysis was determined by two-tailed *t*-test, with the exact *p*-values marked in the graph. **D** Suppression of *Legionella*-infection-caused Rab33b PR-ubiquitination in

HEK293T cells by MavL. HEK293T cells, transfected with Flag-Rab33b, antibody receptor FCγRII, and GFP-MavL (or empty vector), were infected with *L. pneumophila*. At 2-h post infection, the HEK293T cells were lysed and the modification of Rab33b was analyzed after anti-Flag IP. Expression of MavL and Rab33b is shown, with tubulin as a loading control. Experiments were performed three times independently with similar results. **E** Suppression of SdeA-mediated PR-ubiquitination by MavL. HEK293T cells were transfected with Flag-SdeA (or empty vector), GFP-MavL (or empty vector), and HA-Ub^AA. The PR-ubiquitination level in the cell lysate was probed by immunoblotting against HA-tag. Expression of SdeA and MavL is shown, with tubulin as a loading control. Experiments were performed three times independently with similar results. **F** MavL rescues yeast toxicity caused by wild-type SdeA and SdeA PDE mutant. SdeA and mutant were expressed in yeast under the galactose-inducible promotor with MavL or its mutant expressed constitutively. Expression of SdeA and MavL were shown by immunoblotting, with PGK as a loading control. Experiments were performed three times independently with similar results. **G** SidE-mediated noncanonical ubiquitination and its regulation.

E2 enzyme consists of two domains, the RWD domain and UBC domain[32], we investigated which of these is responsible for interaction with MavL. To this end, we cloned these two domains separately with an N-terminal Flag-tag and performed co-IP with HA-tagged MavL after transient transfection in HEK293T cells. We found that the RWD domain (UBE2Q1$_{38-177}$), but not the UBC domain (UBE2Q1$_{214-422}$), interacted with MavL (Supplementary Fig. 2A). Interestingly, MavL$_{41-455}$

lost UBE2Q1 binding, indicating that that the deleted N-terminal segment of MavL is indispensable for UBE2Q1 interaction (Supplementary Fig. 2A). Together, our data showed that the interaction between MavL and UBE2Q1 is dominated by the N-terminal 40-residue segment of MavL and the RWD domain of UBE2Q1. As we could not detect any ADP-ribosylation on UBE2Q1 (Supplementary Fig. 2B, C) either in vivo or in vitro, we were unable to establish if this E2 enzyme is a potential

substrate of MavL. Further study is required to gain deeper insights into the biological outcome of this interaction.

### Substrate specificity of MavL

To better understand the specificity of MavL, we first compared the activity of MavL to that of ARH1 and Larg1. ARH1 belongs to the (ADP-ribosyl)hydrolase (ARH) family, evolutionarily distinct from MavL, and is the only eukaryotic (ADP-ribosyl)hydrolase known to remove MARylation on arginine[33,34]. Unlike MavL, ARH1 exhibited weak (ADP-ribosyl)hydrolase activity towards ADPR-Ub (Fig. 3A), detectable only with a high enzyme concentration (10 μM). Likewise, the recently

reported *Legionella* macrodomain enzyme, Larg1, also exhibited weak (ADP-ribosyl)hydrolase activity towards ADPR-Ub apparent only upon overnight incubation with a high enzyme concentration (Fig. 3A).

Next, we tested if Ub binding is required for MavL activity, we generated a series of Ub mutants (Q40E, Q40L, E51K, E51L, and D52L) with single residue mutations near R42. Through ITC experiments (Supplementary Fig. 3), we found that $Ub^{Q40E}$ and $Ub^{E51K}$ lost binding to MavL, while $Ub^{Q40L}$, $Ub^{E51L}$, and $Ub^{D52L}$ retained binding. $Ub^{Q40L}$ binds to MavL with a $K_d$ of 87.6 μM, similar to that of Ub to MavL (Supplementary Fig. 3B). However, we cannot accurately determine the $K_d$ values for $Ub^{E51L}$ and $Ub^{D52L}$ binding to MavL despite a clear binding

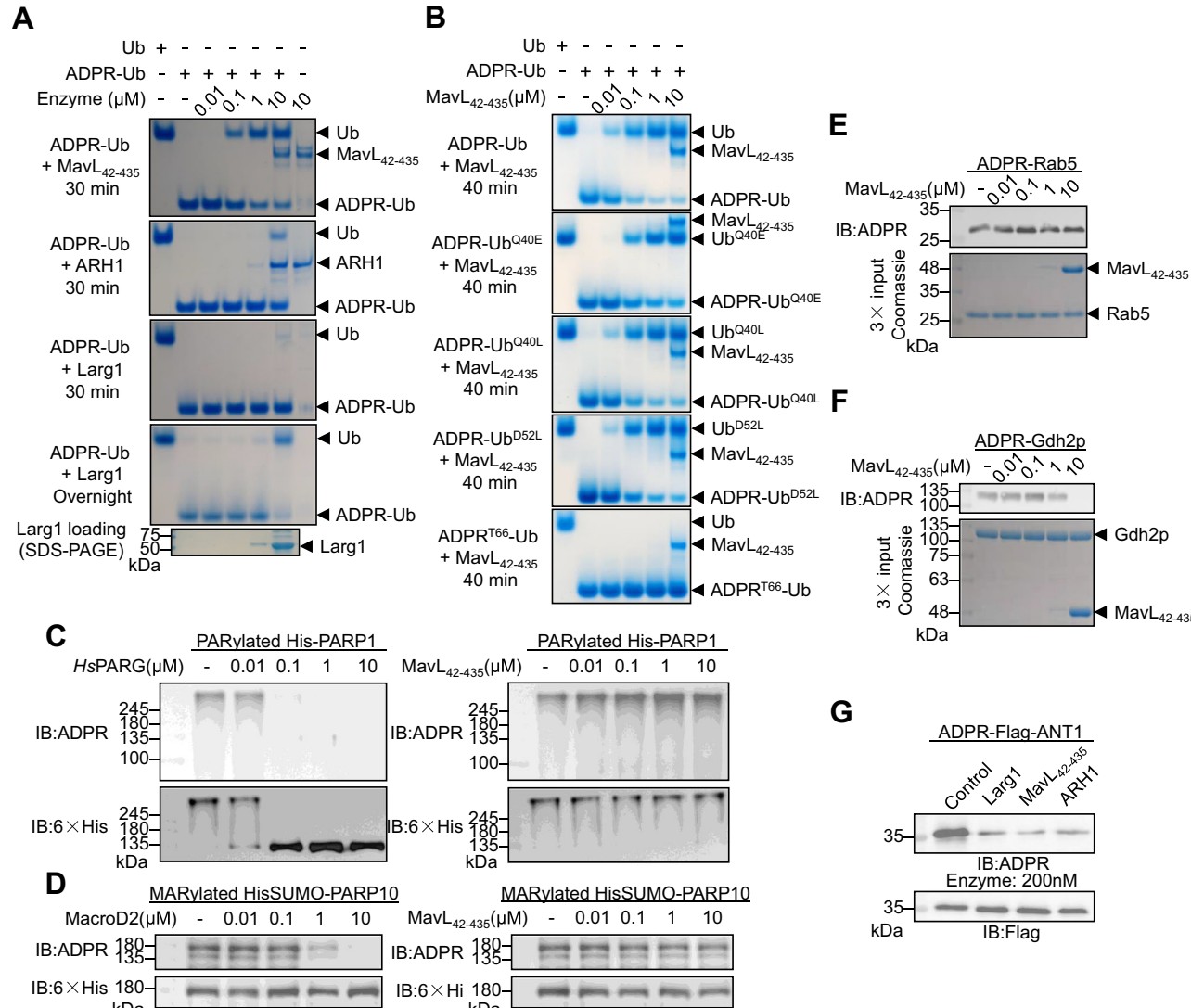

**Fig. 3 | Specificity of MavL activity. A** Comparison of (ADP-ribosyl)hydrolysis of ADPR-Ub by MavL, ARH1, and Larg1. Reactions were analyzed by native PAGE and visualized by Coomassie Blue staining. Controls of Ub, ADPR-Ub, and enzymes alone were included. Experiments were performed three times independently with similar results. **B** Comparison of (ADP-ribosyl)hydrolysis of ADPR-Ub variants by MavL. Reactions were analyzed by native PAGE and visualized by Coomassie Blue staining. Controls of Ub and ADPR-Ub were included. Experiments were performed three times independently with similar results. **C** In vitro de-PARylation assay for MavL. Purified His-tagged PARP1 was auto-PARylated and incubated with varying concentrations of *Hs*PARG (as a positive control) and MavL. Reactions were immunoblotted against ADPR to show the change in ADP-ribosylation level. Loading of the reactions is shown by immunoblotting against 6 × His-tag. Experiments were performed three times independently with similar results. **D** In vitro de-

MARylation assay for MavL. Purified HisSUMO-tagged PARP10 was auto-MARylated and incubated with varying concentrations of MacroD2 (as a positive control) and MavL. Reactions were immunoblotted against ADPR to show the change in ADP-ribosylation level. Loading of the reactions is shown by immunoblotting against 6 × His-tag. Experiments were performed three times independently with similar results. **E–G** Specificity of MavL towards other arginine MARylations. Purified MARylated Rab5 (**E**) and MARylated Gdh2p (**F**) were incubated with varying concentrations of MavL, whereas immunoprecipitated MARylated ANT1 (**G**) was incubated with Larg1, MavL, and ARH1. ADP-ribosylation states of the reactions were probed by immunoblotting against ADPR. Loading of the reactions was visualized by Coomassie Blue staining for Gdh2p and Rab5 and by immunoblotting against Flag-tag for ANT1. Experiments were performed three times independently with similar results.

profile (Supplementary Fig. 3D, E), which we attribute to the relatively small heat changes during titration compared to the baseline signal. We then tested if we could produce ADPR-Ub substrates with these mutants and found that Ub[E51K] and Ub[E51L] cannot be MARylated by SdeA[mART] (Supplementary Fig. 3F). With the remaining mutant substrates, we found that MavL exhibited similar (ADP-ribosyl)hydrolase activity towards ADPR-Ub[Q40L] and ADPR-Ub[D52L] compared to ADPR-Ub (Fig. 3B), whereas lower activity was observed towards ADPR-Ub[Q40E], with no noticeable activity detected at 10 nM MavL (Fig. 3B). These results suggest that Ub binding is indeed needed for optimal activity of MavL, thus reflecting a protein-level specificity of MavL towards Ub. On the other hand, MavL failed to remove MARylation on T66 of Ub (Fig. 3B) introduced by the CteC effector from *Chromobacterium violaceum*[35], suggesting that the (ADP-ribosyl)hydrolase activity of MavL is not solely dependent on Ub recognition.

We then tested if MavL as a macrodomain enzyme could perform the same (ADP-ribosyl)hydrolysis reactions catalyzed by the known macrodomains. Among all macrodomain classes, the ALC1-like class uses a lysine-dependent hydrolysis mechanism[36], which seemed unlikely in the case of MavL based on its structure (discussed later). Therefore, we compared the activity of MavL to that of MacroD-type or PARG-like (ADP-ribosyl)hydrolases. To this end, we used MARylated PARP10 and poly-ADP-ribosylated (PARylated) PARP1[37] as substrates, with MacroD2 and *Hs*PARG, respectively, as positive controls. As expected, (ADP-ribosyl)hydrolase activities were clearly observed in MacroD2 and *Hs*PARG (Fig. 3C, D), whereas MavL did not show any detectable (ADP-ribosyl)hydrolase activities towards these substrates (Fig. 3C, D), indicating inability to cleave ADPR from protein carboxylates and the *O*-glycosidic bond in poly-ADPR chains.

Currently, most studies on (ADP-ribosyl)hydrolases have focused mainly on the residue-level specificities of these enzymes. Therefore, we probed MavL's ability to remove MARylation from arginine in the context of different proteins. To this end, we generated MARylated Rab5 (by *Pseudomonas aeruginosa* mART ExoS, in the presence of mammalian 14-3-3 ζ protein)[38], MARylated yeast glutamate dehydrogenase 2 (Gdh2p) (by *Legionella* mART Lart1)[5], and MARylated ANT1 (by *Legionella* mART Ceg3)[4,9]. While not showing any noticeable (ADP-ribosyl)hydrolase activity towards Rab5, removal of Gdh2p ADP-ribosylation was detected when the substrate was incubated with 10 μM MavL (Fig. 3E, F), showing that MavL can potentially recognize and process arginine MARylation on other substrates. However, MavL can only hydrolyze MARylated Gdh2p at this high concentration, suggesting there may be an inherent preference for MavL at the level of the protein target not just the arginine side chain. Unexpectedly, MavL, ARH1, and Larg1 exhibited similar (ADP-ribosyl)hydrolase activity towards immunoprecipitated MARylated ANT1 (the known substrate for Larg1) (Fig. 3G) under our assay conditions. In cells, ANTs are located across the inner mitochondrial membrane, with the ADP-ribosylation site facing the mitochondrial matrix[4,9]. It has been shown that ARH1 cannot remove ADP-ribosylation of ANT2 localized in mitochondria[9], indicating that the ability of Larg1 to access ANTs within mitochondria may play a critical role in the cellular context. On the other hand, MavL was found to localize throughout cytosol except for nucleus[29], showing that it is unlikely to target ANTs under physiological conditions.

## Structural basis of substrate recognition by MavL

To gather structural insights into the catalytic mechanism, we set out to crystallize MavL and its complexes, starting with apo-MavL for which we obtained a higher resolution structure (2.17 Å) compared to the published one (PDB: 6OMI, 2.69 Å). Our apo-MavL crystals belong to space group $P1\,2_1\,1$, with eight MavL molecules in the asymmetric unit (Supplementary Fig. 4A and Supplementary Table 1). Despite several attempts, we were unable to crystallize a non-covalent complex of MavL with Ub, presumably due to the modest binding affinity

between the two. We therefore resorted to MavL-UbVME covalent adduct, which we managed to crystallize and solve its structure to 2.19 Å (Supplementary Table 1). This complex crystallized in space group $P3_1\,2\,1$ as a crystallographic dimer of two MavL-UbVME in one asymmetric unit (Supplementary Fig. 4B). In the final refined structure, the Ub chain covalently linked to one MavL protomer protrudes away from its expected binding partner only to make contacts with the other MavL chain in the asymmetric unit (Supplementary Fig. 4C, D). In this arrangement, the two MavL molecules, while sharing extensive intermolecular contacts on one face, create a concave surface to hold the intermingled Ub molecules.

Unexpectedly, the R42 of neither Ub in the asymmetric unit was positioned in the ADPR-binding site (discussed below), suggesting that the structure may not represent the catalytically poised state of enzyme-substrate complex. This observation could be attributed to crystal packing or the covalent tether linking MavL to UbVME or the absence of the ADPR moiety in the Ub ligand. Nevertheless, we found an interface in which a MavL loop adjoining the active site, spanning Q103 to T115, forms a series of interactions with Ub, including R42 (Supplementary Fig. 4E). Mutagenesis on this loop showed that F105 and E107 play important roles in MavL activity (Supplementary Fig. 4F). Additionally, the MavL-UbVME covalent complex exhibited similar catalytic activity compared to MavL (Supplementary Fig. 4G), suggesting that this UbVME adduct may not represent an active-like conformation in solution. This is further supported by size-exclusion chromatography coupled with small-angle X-ray scattering (SEC-SAXS, Supplementary Table 2), where we found that in solution, MavL-UbVME is in a monomeric state with an overall elongated shape that does not suggest a biologically meaningful MavL-Ub interface (Supplementary Fig. 6).

We then tried to capture MavL in its ADPR-bound form through co-crystallization of MavL with ADPR and soaking MavL-UbVME crystals with ADPR. While crystals obtained from the co-crystallization trials turned out to be those of apo-MavL, we observed electron density for ADPR in one of the MavL-UbVME subunits from ADPR-soaks and managed to place ADPR within the electron density (Supplementary Fig. 4D and Supplementary Table 1). Inspection of the structure revealed the adenine base of ADPR coordinating between H142 and Y265 of MavL through π-π stacking (Supplementary Fig. 4D). In the other MavL-UbVME complex where ADPR was absent, R370 from the neighboring MavL molecule inserted itself in between the same two His-Tyr side chains (Supplementary Fig. 4D). Interestingly, the same R370-involving protomer:protomer interaction was observed in MavL crystals obtained from co-crystallization trials that failed to yield bound ADPR (Supplementary Fig. 4D). Thus, it appears that the arginine insertion contributes to packing interactions outcompeting ADPR binding. Mutation of R370 to alanine, while retaining normal (ADP-ribosyl)hydrolase activity (Fig. 4B), led to successful capture of ADPR-bound MavL, with well-defined electron density guiding unambiguous placement of the ligand (Fig. 4A and Supplementary Table 1).

The ADPR moiety interacts with multiple residues in MavL. As mentioned before, H142 and Y265 interact with the adenosine via π-π stacking, yet a commonly present aspartate in other macrodomains stabilizing the N6 atom of adenosine[22] is absent in MavL. Furthermore, the proximal ribose is held by interactions with the backbone of G221 and P264 and the side chains of T224 and K236. The di-phosphate mainly interacts with the backbone atoms of the loop containing residues from G223 to F227 (referred to here as the di-phosphate binding loop, loop 1): specifically, the α-phosphate with C226 and D332 and the β-phosphate with G223, G225, and F227. Another loop from D315 to D323 (referred to here as the catalytic loop, loop 2) is responsible for the distal ribose binding in MavL, with D315, N322, and D323 directly forming hydrogen bonds with hydroxyl groups on the C3″, C2″ and C3″, and C1″ atoms, respectively (Fig. 4A). In addition, D333 forms hydrogen bonds to the hydroxyl group on the C2″ atom.

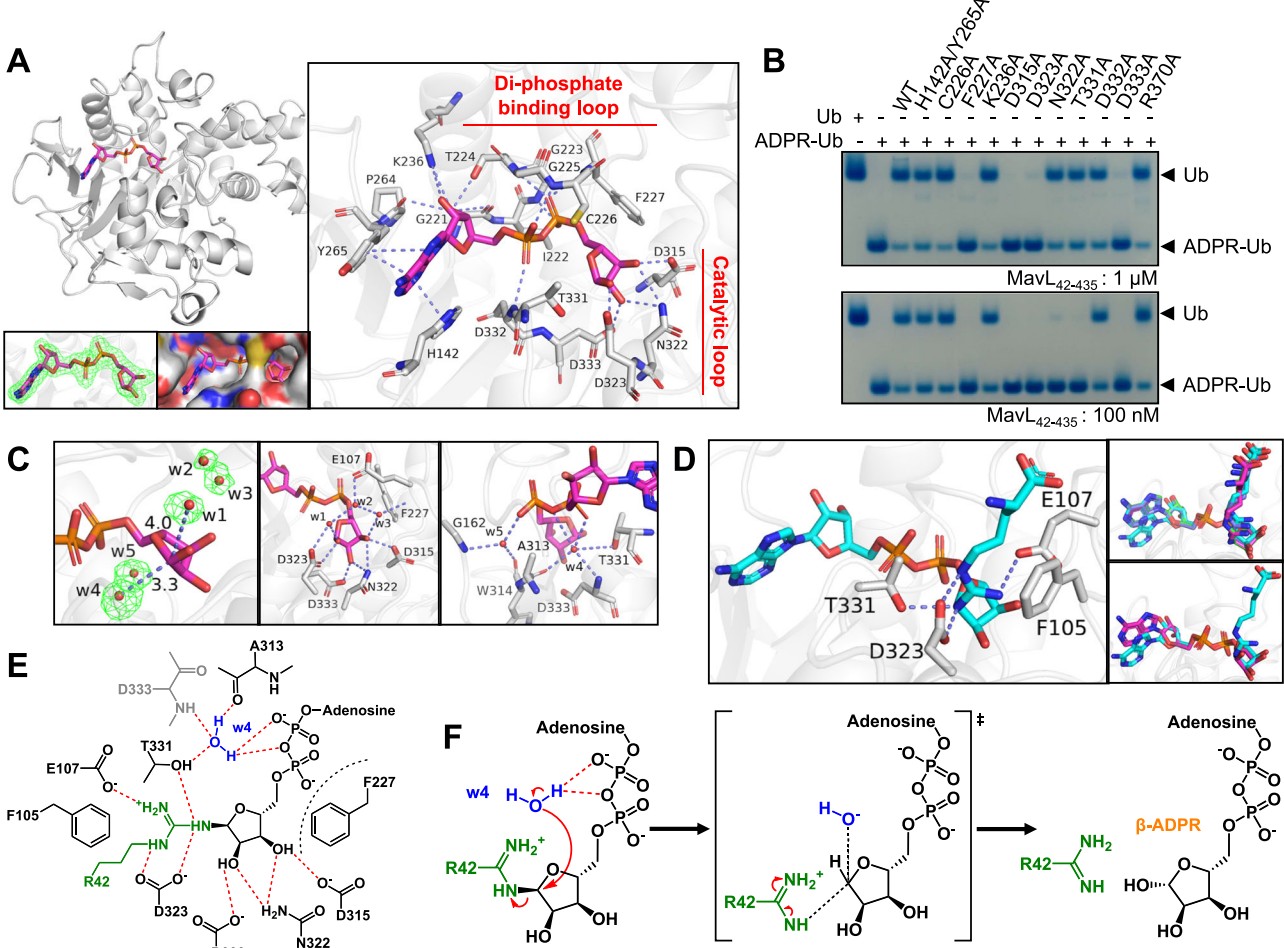

**Fig. 4 | Mechanism of MavL-mediated (ADP-ribosyl)hydrolysis. A** ADPR binding in MavL. Top left figure: Overall structure of ADPR-bound MavL^R370A. Bottom left figure: F_o-F_c map (contour = 3σ) showing the electron density of ADPR. Bottom middle figure: ADPR-binding pocket in MavL. Right figure: ADPR interactions in MavL. The di-phosphate binding loop and the catalytic loop of MavL are marked. **B** Native PAGE showing (ADP-ribosyl)hydrolase activity of MavL mutants based on the ADPR interactions observed in (**A**). Reactions were visualized by Coomassie Blue staining. Experiments were performed three times independently with similar results. **C** Water molecules around the distal ribose in ADPR-bound MavL^R370A. Left figure: Five water molecules (w1 to w5) are shown as red spheres. The w1-C1″ and w4-C1″ distances are marked. Electron density of these water molecules in the F_o-F_c map is shown (contour = 3σ). Middle figure: Interactions of three waters above the distal ribose (w1, w2, and w3) with ADPR and MavL. Right figure: Interactions of two waters below the distal ribose (w4 and w5) with ADPR and MavL. **D** Docking of Arg-ADPR to MavL. Left figure: Top docking model suggests arginine positioning by F105, E107, D323, and T331. Top right figure: Superposition of Arg-ADPR from top five docking models suggests similar Arg-ADPR conformations. Bottom right figure: Superposition of Arg-ADPR from the top docking model and ADPR from ADPR-bound MavL^R370A. Note that the orientations of ADPR are similar. **E** Illustration of substrate interactions in MavL active site, with w4 being the proposed catalytic water. **F** Proposed substrate-assisted S_N2 (ADP-ribosyl)hydrolysis mechanism by MavL.

The side chain of F227 helps position the ribose by providing a steric block from one end. Mutations of D315, D323, D333, or F227, to alanine resulted in a significant loss of activities of MavL (Fig. 4B), whereas N322A and T331A mutants only exhibited activity loss with low enzymatic concentrations.

From the same structure, we observed five ordered waters around the distal ribose, three of them (w1, w2, and w3) above and two (w4 and w5) below the ribose ring (Fig. 4C). Among the waters above the ribose, w1 links D323 to the hydroxyl group on the C1″ atom via hydrogen bonding (Fig. 4C). w2 and w3 form hydrogen bonds to E107 through a water bridge. On the other hand, the substrate-associated w4 is bound to the phosphate groups of the ADPR and further stabilized by the backbone of A313 and D333, and the side chain of T331 (Fig. 4C). This water is the closest to C1″ atom (with a distance of 3.3 Å) (Fig. 4C); accordingly, we propose it to be the catalytic water and its possible activation by the α-phosphate of ADPR for nucleophilic attack at the C1″ center following an S_N2 mechanism[36,39]. To further investigate the arginine recognition by MavL, we docked MARylated arginine with MavL (Fig. 4D) using RosettaLigand[40]. The top five docked structures

have similar MARylated arginine poses that agree well with ADPR orientation from the crystal structure. The docked structure suggests that arginine is coordinated by E107, D323, and T331, with F105 interacting with the aliphatic portion of the side chain. Interestingly, F105 and E107 belong to an insertion loop (see below) that participates in crystal packing of the MavL-UbVME complex (Supplementary Fig. 4E). Despite the lack of direct interactions between ADPR and these two residues, mutation of either F105 or E107 causes noticeable loss of MavL activity (Supplementary Fig. 4F), supporting the docking results. Thus, based on structural and docking results, we propose a substrate-assisted S_N2 catalytic mechanism for MavL. It is worth noting that the α-ADPR captured in the crystal structure likely mimics a substrate-bound state, as a previous crystal structure (PDB: 6B7O) confirmed the α-glycosidic linkage in ADPR-Ub[41]. In this reaction, MARylated arginine first inserts into the active site of MavL, with the distal ribose positioned by F227, D315, N322, and D333, and the arginine residue stabilized by F105, E107, D323, and T331 (Fig. 4E). Next, the di-phosphate-coordinating water activated by the α-phosphate performs nucleophilic attack from below the ribose ring, releasing neutral R42

through a dissociative transition state and generating β-ADPR (Fig. 4F). A similar mechanism has been described for MarcoD-type (ADP-ribosyl)hydrolases, where catalytic water molecules at similar positions were found in several α-ADPR-bound structures[36].

## DUF4804 as a class of Arg-specific macrodomain-type (ADP-ribosyl)hydrolases

Next, we sought to acquire some insights into MavL from the sequence and structural perspectives. As mentioned before, the ALC1-like class employs a lysine-dependent (ADP-ribosyl)hydrolysis mechanism involving a ring-opening Schiff base intermediate[36]. Given no conserved sequence motif has been reported for this class, we compared ADPR-bound structures of MavL, TARG1[42] (PDB: 4J5S), and DarG[24] (PDB: 5M3E). Superposition of ADPR moieties in these structures clearly showed the location of the catalytic lysine residues in TARG1 (K84) and DarG (K80), whereas MavL has an aspartate residue (D315) pointing in a different orientation at the same position (Fig. 5A). Additionally, the overall active site architecture is different in MavL, with the catalytically important residues having no equivalent residues in the ALC1-like macrodomains (Fig. 5A). These structural features suggest that MavL adopts a catalytic mechanism distinct from the ALC1-like class. We therefore focused on the sequence comparison of MavL to MacroD-type and PARG-like macrodomains. Canonical MacroD-type (ADP-ribosyl)hydrolases harbor two signature sequence motifs: Nx(6)GG[V/L/I] and G[V/I/A][Y/F]G present in the catalytic loop and the di-phosphate binding loop, respectively[22,43]. On the other hand, the PARG-like (ADP-ribosyl)hydrolases contain a characteristic GGGx(6–8)QEE catalytic motif in their di-phosphate binding loop (in which the glutamate pair is catalytically essential)[44]. Multiple sequence alignment of MavL with these two classes of (ADP-ribosyl)hydrolases revealed that the sequence motif characteristic of either PARG- or MacroD-type is absent in MavL (Fig. 5B, C), indicating its evolutionary departure from the known macrodomain (ADP-ribosyl)hydrolases. In exploring this further, we performed a homology search of MavL using pHMMER[45] and generated a Hidden Markov Model (HMM) profile based on the multiple sequence alignment of MavL and its homologs. The HMM profile suggested that MavL and its homologs possess two signature sequence patterns (Fig. 5D): [V/I/L]xGxGxGx[F/W/Y] in the di-phosphate binding loop and WDxx[S/A]xPGN[D/E][F/Y][F/W/Y] in the catalytic loop. In MavL, F227 belongs to the first motif (a functional equivalent of this is also present in MacroD-type and PARG-like macrodomains), whereas two other catalytically important residues, D315 and D323, are present in the second one (Fig. 5D). Interestingly, a group of uncharacterized proteins, annotated as DUF4804 in the Pfam database[46], shares highly similar sequence patterns (Fig. 5D) as two conserved motifs. Two of the representative proteins in this group, CG2909, and CG3568, are from *Drosophila melanogaster*, indicating potential evolutionary connections between the prokaryotic effector and the eukaryotic proteins. Further sequence analysis reveals that Larg1 is a distant homolog of DUF4804, with its catalytically important residues, F283, D372, N379, and E380[21], found in the conserved sequence motifs (Fig. 5D).

As we observed weak (ADP-ribosyl)hydrolase activities from the two *Drosophila* proteins towards ADPR-Ub, and no activity towards PARylated PARP1 or MARylated PARP10 (Supplementary Fig. 6), we supposed that this new macrodomain class may harbor specificity toward MARylated arginine. To this end, we screened a panel of well-defined MARylated substrates: synthetic peptides carrying an internal ADP-ribosyl modification on either arginine, serine, threonine, cysteine, or a histidine-mimetic residue[47–50] as well as a ssDNA oligonucleotide with MARylated thymine base (at the N6 atom)[51]. MavL, Larg1, and the *Drosophila* proteins showed distinct hydrolytic activities only towards the arginine-MARylated substrate (Fig. 5E), with no detectable processing of the other types of ADP-ribosylation, whereas these substrates were processed efficiently by their respective

standard enzymes serving as positive controls in the same experiment. These results indicate that the macrodomain class defined by DUF4804 indeed share residue-level substrate selectivity for arginine MARylation. Future studies will be required to test if this residue-level selectivity is a common feature across the entire DUF4804 class.

Encouraged by the clearly detectable ADPR-Ub hydrolytic activity of the two metazoan proteins, we solved their crystal structures in the ADPR-bound forms (ADPR-bound CG2909$_{12-498}$ at 2.28 Å and the ADPR-bound CG3568$_{25-508}$ at 2.00 Å, Supplementary Table 1). Superposition of MavL, CG2909, CG3568, and Larg1 in their ADPR-bound forms reveals a highly similar macrodomain arrangement, but with a noticeable variable region (Fig. 5F). The ADPR interactions in both CG2909 (Fig. 5G) and CG3568 (Fig. 5H) are similar to those observed in MavL, with di-phosphate-binding loops and catalytic loops matching the sequence motifs identified from our search (Fig. 5D). In addition, water molecules equivalent to w4 in MavL can be found in both structures with distances of 3.4 Å (in CG2909, Fig. 5G) and 3.3 Å (in CG3568, Fig. 5H), suggesting that this macrodomain class likely employs a similar catalytic mechanism.

## Structural features unique in Arg-specific macrodomains

Topology comparison of CG2909, CG3568, MavL, Larg1, and *Tc*PARG (Fig. 6A) confirmed a similar macrodomain within this new macrodomain class and revealed its similarity to the *Tc*PARG macrodomain despite some connectivity differences of the secondary structural elements. In Arg-specific macrodomains, the core β-strand connectivity appears to be the same, yet some distinct differences are observed: In both CG2909 and CG3568, expansion of the variable regions from β7 to β9 (compared to β4–β5 in MavL) and the additional β12–β13 were observed. Larg1 lacks one core β strand (β8 in MavL) yet harbors a more expanded variable region from β6 to β10. Additionally, two distinct helices, α9, and α10, are only present in Larg1. These topology differences within this macrodomain class may contribute to their individual protein substrate specificity.

Interestingly, the loop-forming interactions with Ub in MavL-UbVME (Supplementary Fig. 4E) appears to represent a common structural feature in this new macrodomain class, shown as β-sheets in the other three proteins (β2–β3–β4 in CG2909 and CG3568, and β2-β3 in Larg1, Fig. 6A, C). This structural feature is not present in *Tc*PARG (Fig. 6B), indicating that it may be an insertion unique for arginine de-MARylation, supported by the crucial roles F105 and E107 play in the catalysis of MavL (Supplementary Fig. 4F). Interestingly, CG2909, CG3568, and Larg1 all harbor a hydrophobic residue (F137 in CG2909, F148 in CG3568, and V77 in Larg1) equivalent to F105 in MavL in their insertions (Fig. 6D). This hydrophobic residue may participate in arginine binding via hydrophobic interaction with the aliphatic portion of the side chain, as suggested by Arg-ADPR docking to MavL (Fig. 4D). On the other hand, E107 in MavL also seems to have functionally equivalent residues in the other three enzymes (S226 in CG2909, E237 in CG3568, and E160 in Larg1, Fig. 6E), even though these residues are not in the insertions. As E107 in MavL likely coordinates the arginine side chain (Fig. 4D), residues at this position in other enzymes may play similar roles.

To gain more insights into the factors contributing to the arginine specificity, we investigated the depth of the residues binding to distal ribose in PARG-like, MacroD-type, and the Arg-specific macrodomains. We find that in general the distal ribose-interacting residues in the four DUF4804 enzymes have greater depth compared to the other two classes (Supplementary Table 3), indicating that these Arg-specific macrodomains adopt a deeper binding pocket for the distal ribose. The increased depth may exclude binding of short side chains and only allow residues with longer side chains to access the catalytic center. In summary, the presence of the hydrophobic residue above the distal ribose and the depth of the binding cavity may collectively contribute to the arginine specificity of this new macrodomain class.

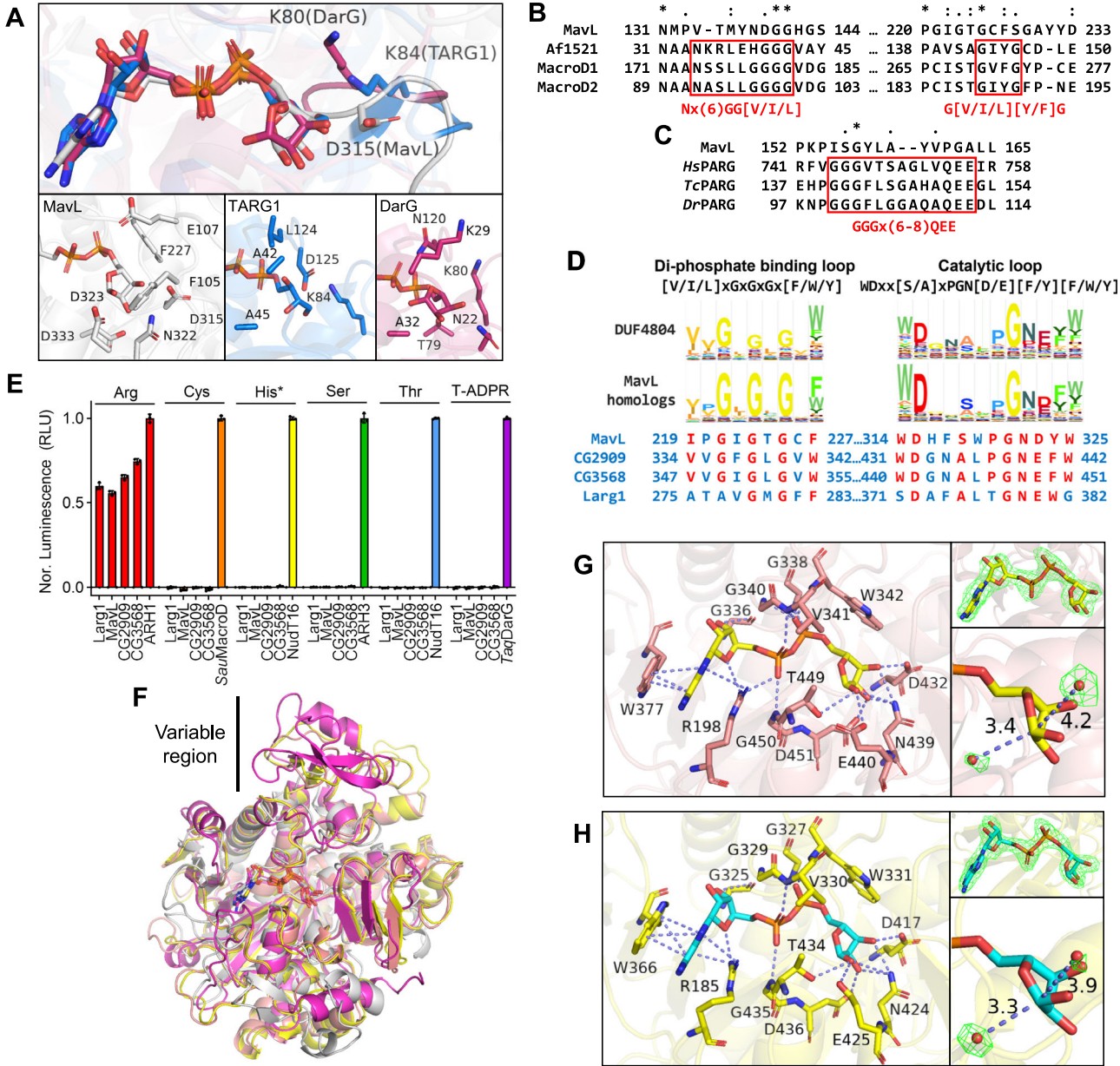

**Fig. 5 | MavL represents a new class of Arg-specific macrodomains present in *Drosophila melanogaster*.** **A** Active site comparison between MavL (gray), TARG1 (blue), and DarG (magenta) suggests MavL has an overall different active site architecture compared to TARG1 and DarG in the ALC1-like class. Note that MavL does not harbor a catalytic lysine residue in its active site. ADPR molecules in these structures are shown as stick models, with colors matching the protein backbone. Phosphate atoms in ADPR are colored in orange. **B**, **C** Multiple sequence alignments of MavL with **B** MacroD-type and **C** PARG-like macrodomains suggest absence of sequence consensus between MavL and either type of macrodomains. **D** Profile Hidden Markov Model logos of MavL homologs and DUF4804. The consensus sequence motifs are shown at the top. Sequences in MavL, CG2909, CG3568, and Larg1 corresponding to the motifs are shown, with the matching residues highlighted in red. **E** Activity screening of Larg1, MavL, CG2909, and CG3568 suggests the substrate preference for arginine MARylation. Well-defined, single MARylated substrates were subjected to hydrolase reaction with (ADP-ribosyl)hydrolases or control enzymes as indicated. The released ADPR was converted to AMP by the

nudix-type phosphodiesterase NudT5 and measured using AMP-Glo assay. Data represent background corrected triplicate measurements. Error bars: standard deviation (SD) of the mean. Controls: Arg, ARH1; Cys, *Staphylococcus aureus* (*Sau*) MacroD; His and Thr, NudT16; Ser, ARH3; T-ADPR, *Thermus aquaticus* (*Taq*) DarG. **F** Superposition of ADPR-bound MavL (gray), CG2909 (salmon), CG3568 (yellow), and Larg1 (magenta, PDB: 7W3S) structures. A variable region is observed upon superposition despite a similar macrodomain arrangement in these enzymes. **G** ADPR binding in CG2909. Left figure: ADPR interactions in CG2909. Top right figure: $F_o$-$F_c$ map (contour = 3σ) showing the electron density of ADPR. Bottom right figure: Two water molecules near C1″ are shown as red spheres. The water-C1″ distances are marked. Electron density of these water molecules in the $F_o$-$F_c$ map is shown (contour = 3σ). **H** ADPR binding in CG3568. Left figure: ADPR interactions in CG3568. Top right figure: $F_o$-$F_c$ map (contour = 3σ) showing the electron density of ADPR. Bottom right figure: Two water molecules near C1″ are shown as red spheres. The water-C1″ distances are marked. Electron density of these water molecules in the $F_o$-$F_c$ map is shown (contour = 3σ).

Although the distal ribose interactions are similar among MavL, CG2909, CG3568, and Larg1 (Fig. 6E), the adenosine interactions are more variable among these enzymes (Fig. 6F). In MavL, the adenosine was stabilized by π-π stacking interaction provided by two aromatic residues (H142 and Y265), whereas in the two *Drosophila* proteins, a π-

cation interaction involving an arginine (R198 in CG2909 and R209 in CG3568) replaces the imidazolium stacking. Larg1, however, exhibited a different adenosine interaction network, with Y134 providing the steric restriction and an aspartate (D351) holding the N6 of adenosine (Fig. 6F). The latter resembles more the aforementioned adenosine-

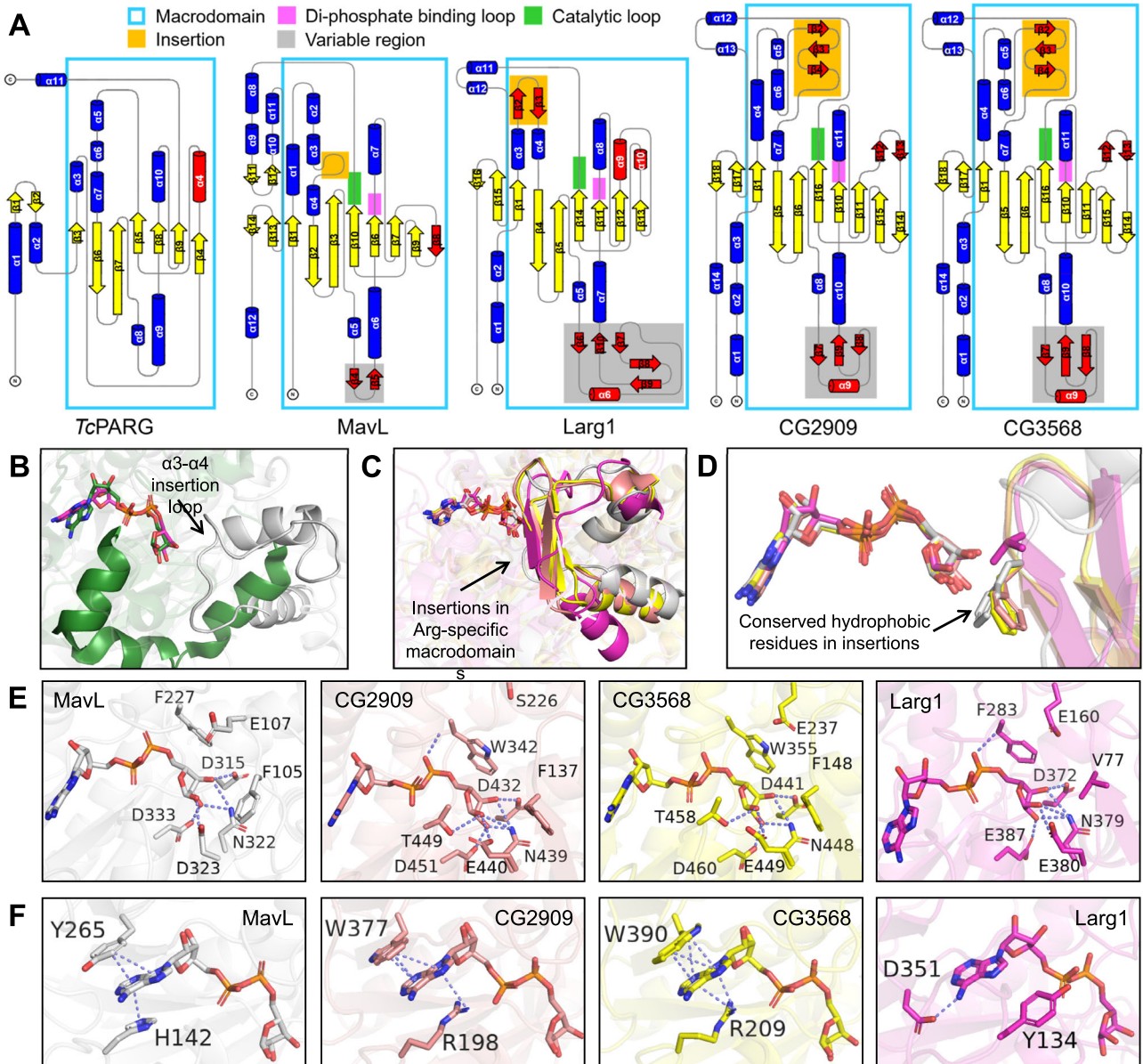

**Fig. 6 | Structural insights into the Arg-specific macrodomains defined by DUF4804. A** Topology diagrams of *Tc*PARG, MavL, CG2909, CG3568, and Larg1. A similar macrodomain core arrangement among these enzymes is observed and highlighted in cyan rectangles. The di-phosphate binding loops, catalytic loops, common structural insertions, and variable regions in Arg-specific macrodomains are highlighted in red, green, yellow, and gray, respectively. **B** Superposition of MavL (gray) and *Tc*PARG (teal, PDB: 3SIG) showing the loop spanning Q103 to T115 in MavL as an insertion that is absent in *Tc*PARG. **C** Superposition of MavL (gray), CG2909 (salmon), CG3568 (yellow), and Larg1 (magenta, PDB: 7W3S) showing the common insertions in these Arg-specific macrodomains. **D** A hydrophobic residue

found at the same position in Arg-specific macrodomain insertions. **E** Interactions of ADPR distal ribose with MavL, CG2909, CG3568, and Larg1 suggest a similar distal ribose binding feature. Note that MavL, CG3568, and Larg1 harbor a glutamate residue (E107 in MavL, E237 in CG3568, and E160 in Larg1) at the same position that does not interact directly with the distal ribose. **F** Interactions of ADPR adenosine with MavL, CG2909, CG3568, and Larg1. Adenosines in MavL, CG2909, and CG3568 are sandwiched between two residues and stabilized by π-π (in MavL) or π-π and π-cation (in CG2909 and CG3568) interactions. In Larg1, the adenosine is positioned by the N6-D351 interaction and the steric restriction provided by Y134.

stabilizing aspartate residue commonly found in other known macrodomains[36], suggesting that Larg1 may have a closer evolutionary connection to other known macrodomain classes.

## Discussion

The noncanonical ubiquitination mediated by the SidE effector family has been shown to be vital for the virulence of *L. pneumophila*, and the presence of additional effectors regulating this pathway ensures a balanced control of this modification[10]. Previous studies have shown that the PR-ubiquitinated host substrates can be reversed by a deubiquitinase-like activity of the effectors DupA and DupB[13,14],

whereas the effectors SidJ and SdjA can polyglutamylate a key catalytic glutamate in the signature R-S-E motif of the SidE mART domain[15–20] to shut off the ubiquitination process at the outset (Fig. 2G). The presence of multiple layers of regulation of the noncanonical ubiquitination pathway underscores the importance of a controlled modification of host targets, as aggressive, irreversible modification of host proteins would be detrimental to the intracellular bacterial lifestyle. Previous studies have shown that ectopic expression of SdeA[mART] alone can be toxic to eukaryotic cells[52], which is likely due to ADPR-Ub being nonfunctional in ubiquitination pathways and the absence of host enzymes to reverse this Ub modification. Studies have also

demonstrated the possibility of ADPR-Ub produced by SdeA$^{mART}$ being first released into the host cytosol before the PDE step of the ubiquitination process, which can lead to ADPR-Ub accumulation if not efficiently processed by the SdeA PDE domain[53,54]. In this scenario, it would be necessary to have a mechanism for producing native Ub from the excess ADPR-Ub. Our results show that Ub can be regenerated by MavL from its ADP-ribosylated form, providing a new regulatory aspect of this noncanonical ubiquitination. The reversal of ADP-ribosylation of Ub is likely to prevent excessive toxicity caused by the accumulation of ADPR-Ub produced by the SidE effector family. Consistent with this notion, expression of MavL, but not its inactive mutant, can significantly reduce yeast cytotoxicity imposed by either the mART domain or the full-length SdeA. Regarding the regulation of SidE family-mediated ubiquitination, DupA and DupB can only rescue the host targets in their native forms, but their activity would still leave Ub in a non-native, modified form as PR-Ub[13,14]. At this point, it is still not clear what would be the metabolic fate of PR-Ub. However, it is worth noting that *Legionella* effectors targeting relevant pathways tend to occur adjacently in the genome[3,12]. In the case of MavL (lpg2526), five other Ub-related effectors have been characterized, including SdjA[15,17] (lpg2508, the polyglutamylase inactivating SidE family mARTs, except for SdeA), DupB[13,14] (lpg2509, the deubiquitinase removing PR-Ub), SdcA[55] (lpg2510, an E3 ligase), SidC[55] (lpg2511, an E3 ligase), and LotC[56,57] (lpg2529, a deubiquitinase). It is therefore tempting to speculate that a gene in proximity to MavL may encode an effector directly hydrolyzing the phosphoribosyl moiety from PR-Ub, or converting PR-Ub into ADPR-Ub for (ADP-ribosyl)hydrolysis by MavL.

Our choice of co-crystallization with UbVME was inspired by abundant precedents of use of this covalent ligand to circumvent low-affinity interaction of deubiquitinases with their Ub product[58]. However, the MavL-UbVME complex structure we captured did not reveal the enzyme-substrate interactions in a catalytically poised state. A number of factors can be responsible for this adventitious capture: not having the ADPR moiety could mean the lack of directing effect of this important substrate feature; tethering the C-terminus of Ub to MavL may restrain the movement of Ub necessary for an induced-fit sort of arrangement at the active site; the relatively low affinity of the ubiquitin product could mean it is readily displaced from its binding site to accommodate more forceful crystallographic contacts, similar to our observation with ADPR soaking experiments with the wild-type enzyme. Even though the MavL-UbVME structure appears to be a crystallographic artifact, the possibility that the interactions we observed may present a pre- or post-catalytic state of MavL-substrate (or product) interactions cannot be ruled out. In the crystal structure of SdeA in complex with Ub bound to its mART domain, instead of R42 (the residue to be ADP-ribosylated) pointing to the active site, R72 of Ub was seen approaching the NAD$^+$ binding site, in an orientation that authors interpreted as representing a pre-catalytic arrangement[54].

Before MavL and the recently discovered Larg1, no macrodomain enzyme was known to harbor arginine de-MARylation activity, and the only enzymes processing arginine MARylations were DraG and ARH1, members of the evolutionarily distinct (ADP-ribosyl)hydrolase (ARH) family[36], which, in contrast to MavL and Larg1, utilize a binuclear metal center in their catalytic mechanism[59]. Our finding of the previously uncharacterized DUF4804 expands the current landscape of this enzyme family. Curiously, DUF4804 appears to have a spotted evolutionary occurrence with only proteins from bacteria, fungi, insects, and other arthropods identified so far. This may indicate a highly specialized function in these organisms, e.g., in the host-pathogen interaction, or an increased evolutionary rate driving diversification and thus hindering identification.

To date, most characterizations of the substrate specificities of MARylation removing macrodomains relied on the residues that are ADP-ribosylated, posing an intriguing question on whether macrodomains have preferences at the protein level. On the residue-level, our activity assays using synthetic MARylated substrate panel showed a clear preference of this new macrodomain class towards hydrolytic reversal of arginine MARylation, with comparable activity among these four enzymes (Fig. 5E). These synthetic peptides represent unfolded structural moieties presenting the MARylated residues to the catalytic action of the enzymes, essentially reporting on residue-level activity. On the other hand, our study also tapped into the protein-level specificity: MavL interacts with Ub and effectively removes ADPR from ADPR-Ub, whereas Larg1, CG2909, and CG3568 exhibited rather weak activity hydrolyzing ADPR-Ub (Fig. 3A and Supplementary Fig. 5). The requirement of protein-level recognition is further demonstrated by the activity difference of MavL towards the MARylated Ub mutants (Fig. 3B). Furthermore, the observation that MavL exhibited weak activity towards MARylated Gdp2h but not MARylated Rab5 (Fig. 3E, F) also supports the speculation that this macrodomain class cares for protein substrates beyond just the MARylated arginine. These results collectively emphasize the context of protein in addition to the MARylated arginine element and suggest a particular evolutionary logic in the construction of these macrodomains. Specifically, the arginine selectivity could have appeared first in the macrodomain scaffold followed by further decoration with substrate-specific elements to achieve specific targeting. For individual arginine-specific macrodomain enzymes, it appears that there is a balance between the recognition of MARylated arginine and the accommodation of local structure near the ADP-ribosylation, which can possibly be achieved by the variable region and the insertion present in the structure (Fig. 6A). Future investigation on the biological substrates for CG2909 and CG3568 would provide deeper insights into the substrate specificity and recognition of this type of macrodomains.

It seems arginine selectivity in DUF4804 can be more complex than what can be learned through inspection of the ADPR-binding site alone. Interestingly, the loop in MavL between α3 and α4 harbors crucial residues for ADPR-Ub hydrolysis (Supplementary Fig. 4F), whereas in *Tc*PARG, the same region is solvent-exposed (Fig. 6B). Larg1, CG2909, and CG3568 while featuring a β-sheet in the same region (Fig. 6C) still harbor hydrophobic residues equivalent to F105 of MavL (Fig. 6D), which likely interacts with the aliphatic portion of the arginine. In addition to the presence of the hydrophobic residues, arginine selectivity can also be attributed to the depth of distal ribose binding pocket (Supplementary Table 3). MARylated arginine may need to be stably positioned at the active site for hydrolysis and the other macrodomains may lack the necessary elements for this purpose. Extrapolating our results from the four proteins, residue-level preference in arginine de-MARylation can be gleaned from a generic peptide substrate. However, this selectivity might have evolved along with the recognition of the protein bearing the modified residue. Further structural investigation of the selectivity determinants by co-crystallization with substrates and substrate-analogs can shed light on this aspect.

## Methods

### Cloning, plasmids, and mutagenesis

For bacterial protein expression, MavL$_{42-435}$, Lart1, Larg1, SdeA$^{mART}$, SdeA$^{mART+PDE}$, Gdp2h, UBE2Q1, CG2909$_{12-498}$ and CG3568$_{25-508}$ were PCR-amplified and cloned into pGEX-6P-1 vector (GE Healthcare). ARH1 in pET41a was a kind gift from Prof. Paul Hergenrother. MacroD2 and *Hs*PARG$_{446-966}$ in pGEX-4T-1, PARP1 in pET-28b, and PARP10 in pET-HisSUMO were kind gifts from Prof. Michael Cohen. CteC in pGEX-6P-1 was a kind gift from Prof. Yongqun Zhu. CG2909 and CG3568 in pGEX-6P-1 were kind gifts from Prof. Sokol Todi. For transfection, MavL and UBE2Q1 constructs were PCR-amplified and cloned in pEGFP-C1 (GFP-tag), pFlag-CMV2 (Flag-tag), or pAPH-HA (HA-tag). For the yeast toxicity assay, MavL and its mutants were PCR-amplified and cloned in p425GPD. Site-directed mutagenesis was performed by PCR-

amplifying the plasmid harboring the construct using mutagenic primer pairs. The methylated template plasmids were removed by the addition of DpnI. All the plasmids were verified by Sanger sequencing before further use. Plasmids and primers used in this study are provided in Supplementary Data 1 and Supplementary Data 2, respectively.

## Antibodies

Antibodies used in this study are mentioned in Supplementary Data 1. Primary antibodies used this studies are as follows: Anti-6×His: Proteintech Cat# 66005 (1:10,000); Anti-pan-ADPR reagent: Sigma Cat# MABE1016 (1:2000); Anti-HA: Invitrogen Cat# 26183 (1:10,000); Anti-Flag: Proteintech Cat# 66008 (1:10,000); Anti-β-actin: ABclonal Cat# AC026 (1:10,0000); Anti-β-tubulin: DSHB Cat# E7 (1:10,000); Anti-PGK: ABclonal Cat# ab154613 (1:10,000); Anti-UBE2Q1: Invitrogen Cat# PA5-70599 (1:800); Anti-SdeA, anti-ICDH, anti-GFP, and anti-*L. pneumophila* antibodies were produced by Pocono Rabbit Farm and Laboratory, Canadensis, PA, with dilution factors of 1:1000.

## Recombinant protein expression and purification

Plasmids for PARP1, PARP10, CG2909, and CG3568 expression were transformed into *E. coli* Rosetta(DE3) strain and the other bacterial expression plasmids were transformed into *E. coli* BL21(DE3) strain. The transformed cells were grown in LB media, except for PARP1 and PARP10 expressing cells which were grown in TB media, at 37 °C to an $OD_{600}$ of 0.6–0.8. The protein expression was induced by addition of 0.4 mM isopropyl-1-thio-β-D-galactopyranoside (IPTG) at 18 °C for 16 h. Cells were then collected via centrifugation at $7000 \times g$ for 7 min and resuspended in 1× phosphate buffered saline (PBS) supplemented with 0.4 M KCl (GST binding buffer). The resuspension was passed twice through a French press under 1500 psi, and the cell debris was removed by ultracentrifugation at $100,000 \times g$ at 4 °C for 1 h. The GST-tagged or His-tagged protein in the supernatant were purified on the glutathione resin or the Ni-NTA resin following manufacturer's instruction. For Ub and Ub mutant purification, cells were collected, resuspended in 50 mM Na acetate pH 4.5, and heated at 80 °C in a water bath for 30 min. After ultracentrifugation at $100,000 \times g$ at 4 °C for 1 h, Ub in the supernatant was captured using SP Sepharose (Cytiva) resin and eluted with 50 mM Na acetate pH 4.5 supplemented with 0.3 M NaCl. Purification of UbVME was performed following a previously described protocol[28]. All the proteins were further purified by size-exclusion chromatography and stored in 50 mM Tris-HCl pH 7.4, 50 mM NaCl, and 1 mM DTT. The purity of the protein was monitored by SDS-PAGE.

Except for Ub, all protein concentrations were measured by Nanodrop using calculated protein molecular weight and extinction coefficient. Concentrations of Ub and Ub activity-based probes were measured using the BCA assay kit.

## Proteome-wide identification of ubiquitin interactors in *Legionella pneumophila*

*L. pneumophila* Lp02 was cultured in ACES-buffered yeast extract media supplemented with thymidine (AYET) until $OD_{600}$ reached 3.0–3.5. The cells were collected and resuspended in 1× PBS supplemented with 0.4 M KCl, 1 mM DTT, and 1 mM phenylmethylsulfonyl fluoride (PMSF). The resuspension was passed twice through a French press under 1500 psi and the supernatant was obtained by ultracentrifugation at $100,000 \times g$ at 4 °C for 1 h. The total protein concentration in the lysate was determined using the BCA assay and adjusted to ~7 mg/mL. 100 μL of the lysate was incubated with 10 μg of the HA-tagged ubiquitin activity-based probes, HA-tagged ubiquitin, or buffer at the same volume. These reactions were incubated at 37 °C for 5 h before immunoprecipitation (IP) via anti-HA agarose. The IP product was eluted by boiling with SDS-loading dye and was run on a 4%–12% SDS-PAGE gel for at 180 V for 5 min. The entire protein lane was

excised, and proteins were in-gel digested by trypsin. The digested peptides were resuspended in 0.1% formic acid and analyzed by LC-ESI-MS/MS using the Dionex UltiMate 3000 RSLC nano System coupled to the Q Exactive™ HF Hybrid Quadrupole-Orbitrap Mass Spectrometer (Thermo Fisher Scientific). One complete dataset was collected ($n = 1$). The raw data files and result files were uploaded to Mass Spectrometry Interactive Virtual Environment (MassIVE) database (identifier: MSV000093623).

For protein identification, the raw data were processed by software MaxQuant 1.6.17.0 with the *Legionella pneumophila* reference proteome (Uniprot proteome ID: UP000000609). The search was performed with the main search peptide tolerance set as 10 ppm. Carbamidomethyl cysteine was set as the fixed modification, and oxidized methionine and acetyl N-terminus were set as variable modifications. Protein false discovery rate was set as 0.01.

## Complex formation, crystallization, and data processing

To generate MavL-UbVME complex, UbVME and MavL$_{42-435}$ at 1:1 molar ratio were co-incubated overnight at room temperature. The resulting reaction was buffer-exchanged into 50 mM Tris-HCl pH 8.0 and purified using anion exchange chromatography using Mono Q column (Cytiva) with 0–400 mM NaCl gradient. The purified MavL-UbVME was further purified by size-exclusion chromatography and stored in 50 mM Tris-HCl pH 7.4, 50 mM NaCl, and 1 mM dithiothreitol (DTT).

To generate the ADPR complex of MavL$^{R370A}$, ADPR-bound CG2909$_{12-498}$, and ADPR-bound CG3568$_{25-508}$, the ADP-ribose solid was dissolved in 50 mM Tris-HCl pH 7.4, 50 mM NaCl and 1 mM DTT and added to the protein solution with a final concentration of 5 mM. The mixture was incubated on ice overnight to allow complex formation.

Apo MavL$_{42-435}$ was concentrated to 20 mg/mL in protein storage buffer and used for initial crystal screening by hanging drop vapor diffusion method at 20 °C. Several hits were observed in the PEG-Ion screen (Hampton Research). These conditions were optimized by altering salt and precipitant concentrations. Eventually, MavL$_{42-435}$ crystals were obtained from the condition containing 0.1 M sodium citrate and 18% PEG3350 with 1:1 ratio of protein solution to reservoir solution. Crystals were observed after 7 days at 20 °C by hanging drop vapor diffusion in this condition and harvested without cryoprotectant. A complete dataset was collected from a single crystal at the Advanced Photon Source (APS) at Argonne National Laboratories on the NE-CAT 24-ID-E beamline (λ = 0.9792 Å). Initial data were indexed and scaled using XDS at 2.17 Å in $P1\,2_1\,1$ space group. Molecular replacement was performed using the program PHASER in the PHENIX suite with the existing MavL structure (PDB: 6OMI) as a template. The structure went through multiple rounds of refinement using COOT and PHENIX suite to generate a final structure. In the final structure, electron density for residues 411 to 435 was not observed. Even though some unsolved electron density observed may be created by the C-terminus of MavL, we could not place any residue with confidence. Hence, these unsolved regions were not modeled. The structure was validated by MolProbity and deposited in the Protein Data Bank (PDB: 8DMP).

MavL$_{42-435}$-UbVME was concentrated to 20 mg/mL in protein storage buffer and used for initial crystal screening by hanging drop vapor diffusion at 20 °C. After 7 days, crystals were seen in 0.2 M sodium citrate with 20% PEG3350 and 0.2 M ammonium citrate with 20% PEG3350 from the PEG-Ion screen (Hampton Research) with 1:1 ratio of protein solution to reservoir solution. Single crystals were looped directly from the initial screen, without cryo-protectant, for data collection. A complete dataset was collected from a single crystal from the condition containing 0.2 M sodium citrate with 20% PEG3350 at the Advanced Photon Source (APS) at Argonne National Laboratories on the NE-CAT 24-ID-E beamline (λ = 0.9792 Å). Initial data were processed and scaled using XDS at 2.195 Å in $P3_1\,2\,1$ space group.

Molecular replacement was done using the program PHASER in the PHENIX suite with existing MavL structure (PDB: 6OMI) and ubiquitin structure (PDB: 1UBQ) as templates. Like apo MavL, residues 411–435 were not modeled due to lack of clear electron density. The structure went through multiple rounds of refinement using COOT and PHENIX suite to generate a final structure. The structure was validated by MolProbity and deposited in the Protein Data Bank (PDB: 8DMQ).

The MavL$_{42-435}$-UbVME crystals used for ADP-ribose-soaking were obtained by manually setting up hanging drops in the condition containing 0.2 M sodium citrate and 20% PEG3350 with 1:1 ratio of protein solution to reservoir solution. After 45 days of setting up drops, crystals were observed. To perform ADP-ribose-soaking, 0.2 μL of 50 mM ADP-ribose dissolved in the reservoir solution was introduced to the 2 μL crystallization drops for soaking, with the final ADP-ribose concentration of 5 mM. The soaking was performed overnight at 20 °C before the crystals were looped for data collection. No cryo-protectant was used when harvesting crystals. A complete dataset was collected from a single crystal at Stanford Synchrotron Radiation Light Source (SSRL) at SLAC National Accelerator Laboratory on the BL12-2 beamline (λ = 0.9795 Å). Initial data were processed and scaled using XDS at 2.15 Å in $P3_1 2 1$ space group. Molecular replacement was performed using the solved MavL$_{42-435}$-UbVME structure. The structure went through multiple rounds of model building and refinement using COOT and PHENIX, respectively, to generate a final structure. The structure was validated by MolProbity and deposited in the Protein Data Bank (PDB: 8DMS).

The ADPR-bound MavL$_{42-435}$$^{R370A}$ was set up for crystal screening with 20 mg/mL protein and 5 mM ADPR by hanging drop vapor diffusion method at 20 °C. After 24 h, several hits were observed in the PEG-Ion screen (Hampton). These conditions were replicated and optimized by altering salt and precipitant concentrations. Eventually, ADPR-bound MavL$_{42-435}$$^{R370A}$ crystals were obtained from the condition containing 0.2 M ammonium sulfate and 20% PEG3350 with 1:1 ratio of protein solution to reservoir solution. Crystals were observed after 24 h at 20 °C by hanging drop vapor diffusion in this condition and harvested without cryo-protectant. A complete dataset was collected from a single crystal at the Advanced Photon Source (APS) at Argonne National Laboratories on the NE-CAT 24-ID-E beamline (λ = 0.9792 Å). Initial data were processed and scaled using XDS at 1.86 Å in $P4_1$ space group. Molecular replacement was performed using the program PHASER in the PHENIX suite with the apo MavL structure as a template. The structure went through multiple rounds of refinement using COOT and PHENIX suite to generate a final structure. Finally, the structure was validated by MolProbity and deposited in the Protein Data Bank (PDB: 8DMR).

The ADPR-bound CG2909$_{12-498}$ was set up for crystal screening with 20 mg/mL protein and 5 mM ADPR by hanging drop vapor diffusion method at 20 °C. After 24 h, crystals were observed in the condition containing 0.1 M ammonium citrate and 12% PEG3350 in the PEG-Ion screen (Hampton). This condition was manually reproduced with 1:1 ratio of protein solution to reservoir solution, and single crystals were looped without cryo-protectant for data collection. A complete dataset was collected from a single crystal at SSRL at SLAC National Accelerator Laboratory on the BL12-2 beamline (λ = 0.9795 Å). Initial data were processed and scaled using HKL3000 at 2.28 Å in $P1 2_1 1$ space group. Molecular replacement was performed using the program PHASER in the PHENIX suite with the predicted AlphaFold structure (Identifier: RE54994p)[60,61] as the template. The structure went through multiple rounds of refinement using COOT and PHENIX suite to generate a final structure. The structure went through multiple rounds of refinement using COOT and PHENIX to generate a final structure. Finally, the structure was validated by MolProbity and deposited in the Protein Data Bank (PDB: 8DMT).

The ADPR-bound CG3568$_{25-508}$ was set up for crystal screening with 20 mg/mL protein and 5 mM ADPR by hanging drop vapor

diffusion method at 20 °C. After 7 days, crystals were observed in the condition containing 0.1 M Tris-HCl pH 8.5, 1 M Li$_2$SO$_4$, and 0.01 M Ni(II)Cl$_2$, in the Structure screen (Molecular Dimension). The condition was replicated and optimized by altering salt and precipitant concentrations. Eventually, ADPR-bound CG3568$_{25-508}$ crystals were obtained from the condition containing 0.1 M Tris-HCl pH 8.5, 1.2 M Li$_2$SO$_4$, and 0.01 M Ni(II)Cl$_2$ with 1:1 ratio of protein to reservoir solution. Crystals were observed after 2 weeks at 20 °C under hanging drop vapor diffusion setup in this condition and harvested without cryo-protectant. A complete dataset was collected from a single crystal at APS at Argonne National Laboratories on the GM-CA 23-ID-D beamline (λ = 0.9793 Å). Initial data were processed and scaled using XDS at 2.0 Å in $P2_1 2_1 2_1$ space group. Molecular replacement was performed using PHASER in the PHENIX suite with the predicted AlphaFold structure (Identifier: AT21585p)[60,61] as the template. In the final structure, electron density for residues 263–268 was not clearly observed. Hence this portion was not modeled. In addition, some unresolved electron density was observed, in which we cannot confidently place any residue. Because the unresolved regions do not seem to affect the overall structure determination of the protein, we left these regions unmodeled. The structure went through multiple rounds of refinement using COOT and PHENIX suite to generate a final structure. Finally, the structure was validated by MolProbity and deposited in the Protein Data Bank (PDB: 8DMU).

## Size-exclusion chromatography coupled with solution small-angle X-ray scattering (SEC-SAXS)

SAXS was performed at BioCAT (beamline 18ID at the Advanced Photon Source, Chicago) with in-line size-exclusion chromatography (SEC-SAXS) to separate sample from aggregates and other contaminants thus ensuring optimal sample quality. The sample was loaded onto a Superdex 75 10/300 Increase column (Cytiva), which was run at 0.6 ml/min by an AKTA Pure FPLC (GE), and the eluate, after it passed through the UV monitor, was flown through the SAXS flow cell. The flow cell consists of a 1.0 mm ID quartz capillary with ~20 μm walls. A coflowing buffer sheath is used to separate sample from the capillary walls, helping prevent radiation damage[62]. Scattering intensity was recorded using an Eiger2 XE 9 M (Dectris) detector which was placed 3.688 m from the sample giving us access to a q-range of 0.027 Å$^{-1}$ to 0.42 Å$^{-1}$. 0.5 s exposures were acquired every 1 s during elution and data were reduced using BioXTAS RAW 2.1.4[63]. Buffer blanks were created by averaging regions flanking the elution peak and subtracted from exposures selected from the elution peak to create the I(q) vs q curves used for subsequent analyses. 3D electron density reconstruction was done using DENSS[64] incorporated in RAW 2.1.4.

## Isothermal titration calorimetry

The ITC experiments measuring the $K_d$ were performed at 25 °C using MicroCal PEAQ-ITC (Malvern Panalytical). Specifically, 200 μM MavL$_{42-435}$ was titrated with 2 mM ADPR or 2 mM Ub (or Ub mutant) in the buffer containing 25 mM HEPES 7.4, 100 mM NaCl, and 1 mM TCEP. Raw data were integrated and analyzed by MicroCal PEAQ-ITC Analysis Software v1.41 (Malvern Panalytical) with a one-binding site model to determine the $K_d$ values.

## Generation of ADP-ribosylated substrates and biochemical assays

All ADP-ribosylation reactions were performed at room temperature using the reaction buffer containing 50 mM Tris-HCl 7.4, 50 mM NaCl, 1 mM DTT, and 1 mM NAD$^+$. To generate ADPR-Gdp2h, 5 μM Gdp2h was incubated with 0.5 μM Lart1 for 2 h. For ADPR-Rab5, 100 μM Rab5 (ΔCAAX) was incubated with 1 μM ExoS and 1 μM mammalian 14-3-3 ζ protein for 2 h. Auto-ADP-ribosylated PARP10 was generated by reacting 5 μM PARP10 in the reaction buffer for 30 min. Auto-ADP-ribosylated PARP1 was generated by reacting 2 μM PARP1 in the

reaction buffer for 30 min. ADPR-Ub was generated by incubating 100 µM Ub with 1 µM SdeA$^{mART}$ (for R42 ADP-ribosylation) or 1 µM CteC (for T66 ADP-ribosylation) for 30 min. After the reaction, ADPR-Gdp2h, ADPR-PARP10, and ADPR-Ub were separated by size-exclusion chromatography to remove the ADP-ribosyl transferases and excessive NAD$^+$. The purified ADP-ribosylated protein was stored in 50 mM Tris-HCl 7.4, 50 mM NaCl, and 1 mM DTT. ADPR-Rab5 reaction was buffer-exchanged into 50 mM Tris-HCl 7.4, 50 mM NaCl, and 1 mM DTT using a concentrator to remove excessive NAD$^+$. ADPR-PARP1 reaction was stopped by adding 30 µM olaparib. ADPR-ANT1 and Ub-PR-Rab33b were generated and purified as previously described[6,21].

To investigate the activity of MavL on Ub variants, 100 µM Ub, PR-Ub, or ADPR-Ub was incubated with 2 µM MavL at 37 °C for 30 min in a buffer containing 50 mM Tris-HCl 7.4, 50 mM NaCl, and 1 mM DTT. Controls containing only Ub, PR-Ub, ADPR-Ub, or MavL were incubated in the same condition. These reactions were run on 10% native PAGE gels at 150 V for 50 min with the running buffer containing 25 mM Tris-Glycine, pH 8.3. The gels were stained with Coomassie blue stain.

To compare the activities of DupA, DupB, and MavL, 100 µM ADPR-Ub was incubated with 1 µM DupA, DupB, or MavL at 37 °C for 30 min in a buffer containing 50 mM Tris-HCl 7.4, 50 mM NaCl, and 1 mM DTT. Controls containing only Ub, PR-Ub, ADPR-Ub, or enzymes were incubated in the same condition. These reactions were analyzed immediately after incubation via native PAGE and SDS-PAGE and stained with Coomassie blue stain or phosphoprotein stain (ABP Biosciences, for detection of PR-Ub in SDS-PAGE gel). The phosphoprotein staining was performed following manufacturer's protocol. To access the deubiquitination of Ub-PR-Rab33b, 2 µM of HA-Ub-PR-His-4 × Flag- Rab33b was incubated with 1 µM DupA, DupB, or MavL at 37 °C for 30 min in a buffer containing 50 mM Tris-HCl 7.4, 50 mM NaCl, and 1 mM DTT. A control without enzyme was included. These reactions were analyzed immediately after incubation via SDS-PAGE and probed with anti-HA and anti-Flag immunoblotting.

For (ADP-ribosyl)hydrolysis of ADPR-Ub, 100 µM of ADPR-Ub was incubated with 0.01 µM to 10 µM MavL, Larg1, CG2909, CG3568, or ARH1. The reaction was performed at 37 °C for 30 min in a buffer containing 50 mM Tris-HCl 7.4, 50 mM NaCl, and 1 mM DTT, with additional 4 mM MgCl$_2$ added for ARH1 reactions. Controls containing only Ub, ADPR-Ub, or enzymes were incubated in the same condition. These reactions were analyzed immediately after incubation via native PAGE and stained with Coomassie blue stain.

For (ADP-ribosyl)hydrolysis of ADPR-ANT2, 2 µM of ADPR-ANT2 was incubated with 1 µM MavL, Larg1, or ARH1. The reaction was performed at 37 °C for 30 min in a buffer containing 50 mM Tris-HCl 7.4, 50 mM NaCl, 4 mM MgCl$_2$, and 1 mM DTT. A control containing only ADPR-ANT2 was incubated in the same condition. These reactions were analyzed immediately after incubation via SDS-PAGE and probed with anti-ADPR immunoblotting.

To compare the activity of MavL mutants, 100 µM ADPR-Ub was incubated MavL or mutants at 37 °C for 30 min in a buffer containing 50 mM Tris-HCl pH 7.4, 50 mM NaCl, and 1 mM DTT. Reactions were carried out under 100 nM or 1 µM MavL. Controls containing only Ub, ADPR-Ub were incubated in the same condition. These reactions were analyzed immediately after incubation via native PAGE and stained with Coomassie blue stain.

For (ADP-ribosyl)hydrolysis of ADPR-Gdp2h and ADPR-Rab5, 2 µM ADPR-Gdp2h or ADPR-Rab5 was incubated with 0.01 µM to 10 µM MavL at 37 °C for 30 min in a buffer containing 50 mM Tris-HCl 7.4, 50 mM NaCl, and 1 mM DTT. These reactions were quenched by the addition of SDS-PAGE loading buffer with boiling at 95 °C for 5 min. The ADP-ribosylation state of the substrates was probed via immunoblotting against ADPR. Loading of the reaction was shown by SDS-PAGE with Coomassie blue staining.

For (ADP-ribosyl)hydrolysis of ADPR-PARP1 and ADPR-PARP10, 2 µM ADPR-PARP1 or ADPR-RAPR10 was incubated with 0.01 µM to 10 µM MavL, HsPARG (for ADPR-PARP1), or MacroD2 (for ADPR-PARP10) at 37 °C for 30 min in a buffer containing 50 mM Tris-HCl 7.4, 50 mM NaCl, and 1 mM DTT. These reactions were quenched by the addition of SDS-PAGE loading buffer with boiling at 95 °C for 5 min. The ADP-ribosylation state of the substrates was probed via immunoblotting against ADPR. Loading of the reaction was shown via immunoblotting against 6 × His-tag on the PARPs.

## AMP-Glo (ADP-ribosyl)hydrolase activity assay
The peptide demodification assay was described earlier[47]. Briefly, peptide concentration for the assay were estimated using absorbance at $\lambda_{260nm}$ with a molar extinction coefficient of 13,400 M$^{-1}$ cm$^{-1}$ for the ADP-ribosyl modification. 8 µM of indicated peptide was demodified by incubation with 1 µM indicated hydrolase (or various concentrations of MavL/Larg1) for 30 min at 30 °C in assay buffer (50 mM Tris-HCl pH 8, 200 mM NaCl, 5 mM MgCl$_2$, 1 mM DTT and 0.15 µM human NUDT5)[65]. Reactions were stopped and analyzed by performing the AMP-Glo™ assay (Promega) according to the manufacturer's protocol. Luminescence was recorded on a SpectraMax M5 plate reader (Molecular Devices) and data analyzed with GraphPad Prism 7. For background subtraction reaction were carried out in absence of hydrolase.

## Transfection, immunoprecipitation, and *L. pneumophila* infection
HEK293T cells (ATCC, Cat# CRL-3216) were cultured in DMEM containing 10% FBS at 37 °C with 5% CO$_2$ until ~70% confluence. Plasmids were transfected using Lipofectamine 3000 following manufacturer's protocol. 24 h following the transfection, the cells were collected and lysed using the ice-cold buffer containing 50 mM Tris-HCl pH 7.4, 150 mM NaCl, and 1% Triton X-100. The cell lysate was cleared by centrifugation at 13,000 × g at 4 °C for 15 min. Immunoprecipitation was performed by incubating the cleared cell lysate with anti-Flag resin at 4 °C overnight. The resin was washed thrice with the cold lysis buffer. The bound protein was eluted using 1× PBS containing 150 µg/mL 3× Flag peptide. The cell lysate and IP products were resolved on SDS-PAGE gels and probed with appropriate antibodies via immunoblotting.

For *L. pneumophila* infection, *L. pneumophila* strains were grown in AYE broth to the post-exponential phase (OD600 = 3.2–4.0). For the *L. pneumophila* growth assay, approximately 4 × 10$^5$ RAW264.7 macrophage cells (ATCC, Cat# TIB-71) per well were seeded into a 24-well plate a day before infection. The next day, *L. pneumophila* was added to the wells at a multiplicity of infection (MOI) of 0.05 for infection. To synchronize the infection, the macrophage monolayers were washed with 1× PBS thrice 2 h after infection. The infected macrophages were incubated at 37 °C with 5% CO$_2$. At each time point, cells were lysed by 0.02% saponin, and dilutions of the lysate were plated onto bacteriological media. The colony-forming units (CFU) were determined from triplicate wells of each strain.

For examining ADPR-Ub level under infections by Lp02Δ*mavL* and complementary strains: HEK293T cells were co-transfected with plasmids expressing the FcγII receptor and 3× HA-Ub. Cells were supplied with fresh media 6 h after transfection. *L. pneumophila* strains were grown to the post-exponential phase (OD$_{600}$ = 3.3–4.0) in AYET media supplemented with 20 µg/mL kanamycin for 14 h at 37 °C. Six hours prior to infection, the expression of Flag-MavL was induced by adding IPTG with a working concentration of 0.3 mM. *L. pneumophila* strains were then opsonized with anti-*L. pneumophila* antibodies for 1 h at 37 °C before infecting HEK293T cells at an MOI of 1:30. Two hours post infection, cells were washed with cold PBS for three times and then lysed in TBS buffer containing 150 mM NaCl, 50 mM Tris-HCl, 1 mM DTT, 1% Triton X-100, 1% SDS, 1× complete protease cocktail, following with 10% sonication for 30 s. After boiling for 10 min, lysates were centrifuged at 20,000 × g for 15 min. Supernatants were collected,

mixed with 6× SDS-loading buffer, resolved on SDS-PAGE gels, and probed with appropriate antibodies via immunoblotting.

For the infection-based Rab33b PR-ubiquitination assay, the HEK293T cells were transfected to express Flag-Rab33b, GFP-MavL (or the D333A mutant), and the FCγRII receptor for 24 h. To probe the ADPR-Ub level under infection condition, the HEK293T cells were transfected to express HA-Ub and the FCγRII receptor for 24 h. These cells were then infected with *L. pneumophila* opsonized by anti-*LpO2* rabbit antibodies. After 2-h infection, the cells were harvested, washed with ice-cold 1× PBS thrice, and lysed with 50 mM Tris-HCl pH 7.4, 150 mM NaCl, and 1% Triton X-100 for further immunoblotting analysis.

### Yeast toxicity assays

MavL or MavL$^{D333A}$ mutant was cloned in p425GPD plasmid and transformed into yeast W303 strains harboring SdeA or SdeA$^{H227A}$ in pYES1NTA (described in ref. 6). Controls of yeast harboring two empty plasmids and empty p425GPD with SdeA or SdeA$^{H227A}$ were included. Yeast grown in appropriate liquid dropout media supplemented with glucose was washed and serially diluted in sterile water and 4 µl of each dilution was spotted onto selective plates either 2% glucose or 2% galactose. The plates were incubated at 30 °C for 3 days before observation.

### Bioinformatics

The docking of Arg-ADPR and MavL was performed using RosettaLigand[40] web server. The topology diagrams of the protein structures were manually drawn using TopDraw software[66] based on initial topology diagrams generated by Pro-origami web server[67]. Homology search was performed using HMMER server[45] with the Hidden Markov Model (HMM) profile generated by the program. The HMM logos of MavL homologs and DUF4804 were generated from the HMM profiles using Skylign web server[68]. The residue depth analysis was performed using the DEPTH web server[69], with the distal ribose-interacting residues manually inspected.

### Quantification and statistics analysis

All biochemical and cell assays were performed in biological triplicates, with representative results presented here. List of identified Ub-interacting effectors is shown in Supplementary Data 3. Crystallographic data collection and refinement statistics are listed in Supplementary Table 1. SEC-SAXS data collection and analysis parameters are listed in in Supplementary Table 2.

### Reporting summary

Further information on research design is available in the Nature Portfolio Reporting Summary linked to this article.

## Data availability

Structural factors and atomic coordinates of MavL$_{42-435}$, MavL$_{42-435}$-UbVME, ADPR-bound MavL$_{42-435}$-UbVME, ADPR-bound MavL$_{42-435}$$^{R370A}$, ADPR-bound CG2909$_{12-498}$, and ADPR-bound CG3568$_{25-508}$ have been deposited to Protein Data Bank with accession codes 8DMP, 8DMQ, 8DMS, 8DMR, 8DMT, and 8DMU. Proteomics dataset identifying Ub-interacting Legionella effectors has been deposited to Mass Spectrometry Interactive Virtual Environment (MassIVE) with the accession code MSV000093623. Other data, including full gels, blots, and raw data used to generate plots are provided in the Source Data file. Source data are provided with this paper.

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

## Acknowledgements

We thank Dr. Uma K. Aryal at Purdue Proteomics Facility in Bindley Bioscience Center for support in LC-MS/MS data collection and data analysis. We thank Prof. Robert Stahelin and Dr. Lan Chen at Purdue University for their assistance with the isothermal titration calorimeter. We acknowledge the use of the Chemical Genomics Facility, a core

facility of Purdue Institute for Drug Discovery, and the NIH-funded Indiana Clinical and Translational Sciences Institute. We thank the staff contacts on NE-CAT beamline 24-ID-E and GM-CA 23-ID-D at the Advanced Photon Source (Argonne National Laboratory) and on beamline BL12-2 at Stanford Synchrotron Radiation Light Source (SLAC National Accelerator Laboratory), for their assistance during X-ray data collection and analysis. This research used resources of the Advanced Photon Source, a U.S. Department of Energy (DOE) Office of Science User Facility operated for the DOE Office of Science by Argonne National Laboratory under Contract No. DE-AC02-06CH11357. We thank Prof. Sokol V. Todi, Wayne State University, for plasmids encoding the Drosophila genes, Daniel Sanderson and Prof. Michael S. Cohen, Oregon Health & Science University, Prof. Paul J. Hergenrother, University of Illinois Urbana-Champaign, and Prof. Yongqun Zhu, Zhejiang University for kindly providing other plasmids. We thank Hugo Minnee at Leiden University for making the MARylated histidine-mimetic peptide. The ubiquitin activity probes were a kind gift from South Bay Bio (San Jose, CA). This study was funded in part by National Institute of Health Grants R01GM126296 (to C.D.), R21AI171709 (to C.D.), R01AI127465 (to Z-Q.L.), and T32AI148103 (to Z.Z.). BioCAT was supported by grant P30 GM138395 from the National Institute of General Medical Sciences of the National Institutes of Health. Work in the J.G.M.R. laboratory was supported by the Medical Research Council (MR/X007472/1). We acknowledge funding from the MRC Centre for Medical Mycology at the University of Exeter (MR/N006364/2 and MR/V033417/1) and the NIHR Exeter Biomedical Research Centre. Additional work may have been undertaken by the University of Exeter Biological Service Unit. The views expressed are those of the authors and not necessarily those of the NIHR or the Department of Social Care. The work in the I.A. laboratory was supported by the Wellcome Trust (210634 and 223107), Biotechnology and Biological Sciences Research Council (BB/R007195/1), Ovarian Cancer Research Alliance (813369) and Cancer Research United Kingdom (C35050/A22284).

## Author contributions

Z.Z. and C.D. conceived and designed the study. J.F. and C.L. performed the cell-based assays probing PR-ubiquitination and Ub ADP-ribosylation. J.F. and Z.Z. performed the Legionella intracellular growth assay. J.G.F.R. performed the AMP-Glo (ADP-ribosyl)hydrolase activity assay. J.V. synthesized and provided the MARylated peptides. Z.Z., J.G.F.R., and C.D. proposed the mechanism. Z.Z. performed all other experiments including LC-MS/MS sample preparation, crystallization, structure determination, biochemical assays, and bioinformatical analysis. Z.Z. wrote the paper with editorial inputs from D.F., I.A., Z.-Q.L., C.D., and all other authors.

## Competing interests

The authors declare no competing interests.
