## [Peer Review File · Nature Communications]

Legionella metaeffector MavL reverses ubiquitin ADP-ribosylation via a conserved arginine-specific macrodomainReviewer #1 (Remarks to the Author):

In the manuscript entitled "Legionella metaeffector MavL reverses ubiquitin ADP-ribosylation via a conserved arginine-specific macrodomain", the authors clearly showed that MavL harbors (ADP-ribosyl)hydrolase activity having substrate preference to ADP-ribosylated ubiquitin, formed by mART activity of SidE family Legionella effectors. They also showed that MavL has metaeffector function against the SidE family effectors but the experimental validation was limited in HEK293T cells producing MavL by transient transfection. Structural analysis of ADPR-bound MavL identified residues forming the ADPR-binding pocket, while that of MavL-Ub is not great because good co-crystals were not available. Moreover, the authors analyzed CG2909 and CG3568 from *Drosophila*, both of which as well as MavL belong to DUF4804 family, and the authors concluded that the DUF4804 family represents a new class of Arg-specific macrodomains harboring (ADP-ribosyl)hydrolase activity.

Major comments

1. (Figure 2 panel BC) To demonstrate that MavL functions as a metaeffector against the SidE family, the authors ectopically expressed MavL or its catalytically-dead derivative in cells. To support the important conclusion of this paper, the experiment should be carried out in more physiological systems, such as infection using MavL catalytically-dead (or KO) and/or MavL overproducing strains.
2. (Figure 2 panel D) The authors should provide the data indicating that (over)production of mART-dead SdeA does not result in yeast toxicity as a control experiment.

Minor comments

3. The entire results of the proteome-wide Ub-interactor screening (Figure 1 panelA) should be presented as a supplemental table, as the data would be good resources.
4. Description of full-length MavL and MavL42-435 is confusing especially in figures. For example, "MavL"s in figure 1 panels B-D should be labeled as "MavL42-435".
5. (Figure 1 panel A) what do the red texts (+HA-Ub, +buffer) mean?
6. (Figure 2 panel F) Although I agree that this is a piece of important data which should be published, this data is nothing to do with the main logic of this figure/paper. Supplemental figure is a proper place to have this panel.
7. (Figure 3 panel A) In Coomassie gels bands of MavL42-435 and ARH1 are readily visible, but not Larg1. It is nice to add a (native or denature) gel image showing that comparable amounts of Larg1 is included in the reactions.
8. (Figure 7 panel D) To achieve proper sequence alignment is one of the most important prerequisites of phylogenetic analysis. I believe that sensible alignments containing remote Larg1 sequence is pretty much hard or even impossible to obtain. How do the authors align sequences and how the alignment is validated? Furthermore, the authors do not provide any details about the phylogenetic analysis. Which evolution model is employed to confer the ML tree and why? No statistical validation of the tree such as bootstrap supports is indicated. The current tree is nonsense and should be removed from the figure.
9. Citations to ref 9 are missing in many places regarding Larg1 in texts.

Reviewer #2 (Remarks to the Author):

In their manuscript "Legionella metaeffector MavL reverses ubiquitin ADP-ribosylation via a conserved arginine-specific macrodomain", Zhang et al. discover and characterize a new class of ADP-ribosyl hydrolase enzymes unrelated to those currently known in humans. *Legionella pneumophila* secretes numerous effector proteins upon infection, some of which have been recently characterized to produce new ubiquitin modifications and linkages. The SidE family of Legionella E3 ligases ubiquitinate substrate proteins using an R42 ADP-ribosylated ubiquitin intermediate that is then conjugated onto target serine residues via a phospho-ribose linkage. The Legionella deubiquitinases DupA and DupB reverse SidE-mediated ubiquitination by removing PR-Ub from substrates. Using a proteomics workflow, Zhang et al. identify the Legionella effector MavL as a ubiquitin-interacting protein, and provide evidence that it restores ADPR-Ub to its native form. They go on to suggest that MavL acts as a metaeffector, controlling the activity of the SidE

ligase family. Following extensive structural characterization, the authors reveal that the arginine-specific macrodomain of MavL represents a novel class in this enzyme family, with structurally-related examples observed in eukaryotes including *Drosophila melanogaster*. This manuscript is mostly well written and the figures are easy to follow. It will be of interest to microbiologists and structural biologists across both the ubiquitin and ADPR fields. While the work is of high interest and quality, there are several issues that should be resolved. Primary among them are MavL's role as a metaeffector and its specificity toward ubiquitin. This manuscript could be a candidate for publication in *Nature Communications* if the following concerns are addressed:

Major Concerns:

1. The first figure describes a mass spectrometry-based approach to finding Ub-interacting proteins in *Legionella*, but there is no indication in the manuscript of where the data will be deposited. These data should be made publicly available following publication.
2. The crystal structure of MavL-UbVME does not accurately represent the enzyme:substrate complex, but the authors have discussed and justified findings as if it does. Based on their data, it is possible that the ubiquitin-bound structure is an artifact of crystallization. It is unclear what is learned from the ubiquitin-bound structure that could not be gleaned from the apo and ADPR-bound structures, and thus the authors should consider removing it from the manuscript entirely if it is indeed an artifact.
 - a. R42 of ubiquitin is positioned far away from the MavL active site, and furthermore almost none of the mutations tested at the ubiquitin interface have an effect on MavL activity. These signs point toward the structure representing a crystallization artifact, which in this reviewer's opinion should not be published as it will lead to future confusion in the field. Alternatively, the visualized ubiquitin-binding interface could be genuine, but distinct from the binding site for hydrolase activity. Have the authors tested any of their MavL mutants for a defect in ubiquitin binding?
3. Several aspects of the authors' proposed enzymatic mechanism are overstated and should either be scaled back or solidified with more supporting data.
 - a. The authors show that both F105 and E107 are crucial for the removal of ADPR from ubiquitin, but the argument that they position the R42 sidechain (aliphatic and charged interactions, respectively) is speculative. Structural modeling of an Arg residue coming off the ADPR would indicate side chain direction and provide stronger support for the authors' statements. Additionally, mutation of these sites indicates their importance for activity, but not necessarily specificity. Have the authors determined if the F105 or E107 mutants are suddenly active toward other forms of ADP ribosylation?
 - b. The authors discuss the ADPR-bound structures as if they are bound to substrate, but in fact they are bound to product. This leads them to claim that they observe a catalytic water molecule in their ADPR-bound MavL structure (w4), but there is no biochemical support for this.
 - i. The T331 sidechain is shown to coordinate w4 in Fig 5C/D but makes no contact to the ADPR molecule. If w4 were the catalytic water, why does substitution at T331 not effect MavL activity determined in Fig 4F?
 - ii. Are these waters also observed in similar positions in the CG2909 and CG3568 ADPR-bound structures?
 - c. Fig 5D shows D323 coordinating the R42/ribose connection, yet the authors also propose that E107 serves this function. Which is it?
 - d. How would the authors explain the lack of an effect from removing the ability of H142 and Y265 to stack with the adenosine ring of ADPR?
4. Regarding MavL's function as a metaeffector, the authors show that cells ectopically expressing MavL are defective for SdeA-mediated ubiquitination following either *Legionella* infection or SdeA ectopic expression. While consistent with this possibility, both of these experiments are far from native conditions. The authors already have a Δ mavL *Legionella* strain; following infection do you observe an increase in either ADPR-Ub, Ub-Rab33b, or total PR-ubiquitination compared to WT *Legionella*? Regarding the interpretation of this assay, have you formally demonstrated that MavL cannot hydrolyze PR-ubiquitinated substrates?
5. The yeast toxicity assay should be repeated with serial dilution plating and western blotting to show expression levels of the various constructs.
6. The BLI curve fits for the MavL:ubiquitin interaction do not look very good. Can the authors explain this?
7. There are several inconsistencies in the reported western blot experiments:
 - a. In Fig 3B, why does the addition of more MavL increase the amount of ADPR detected for

PARYlated His-PARP1? If the amount of MavL is the only thing changing, and MavL is not reactive to this species of ADPR-protein, shouldn't the lanes all look identical?

b. There appears to be a loading issue for the blots presented in Fig 3C. The 6His blot should be consistent but appears to diminish with increasing concentration of MavL.

c. There is inconsistent loading in several other control blots including S4B, S4C.

8. As further support that MavL is specifically a ubiquitin-directed ADPR hydrolase, it would be useful to include ADPR-Ub as a substrate in the luminescence assay presented in Fig 6D for comparison to the ADPR-Arg substrate.

Minor Concerns:

1. In the majority of structure figures, protein subunits are not labeled and coloring is not indicated. This would clarify things and orient the reader better.

a. Additionally, the coloring used to highlight interactions in the CG3568 structure presented in Fig. 7E/F are difficult to discern. The authors should consider color choice for the structure and change the appropriate figures.

2. The authors should read the manuscript carefully before re-submission and remove extra words, check for grammar, correct nomenclature inconsistencies etc.

3. The methods section should include a description of how the native PAGE was performed.

4. Authors show an interaction between UBE2Q1 and MavL but do not explore this further. This seems out of place in this manuscript.

5. It is not clear from the figure legends that "V" denotes empty vector control.

6. The orientation in Fig 5C shows the A313 backbone amide coordinating w4, but the authors show the backbone carbonyl coordinating w4 in Fig 5D.

7. Figures 7D and 7E are presented out of order.

8. Labeling the two loops involved in ADPR binding in the topology diagrams of Fig 7A would be helpful.

9. References to Table S1 should be included in the areas of text that describe each crystal structure.

10. On page 8, there is a description of the conservation of E107 that references Fig 7F. I believe this should be referencing Fig 7E.

11. In the discussion, it is unclear why this work suggests the existence of an additional enzyme that converts PR-Ub to ADPR-Ub. Is it not conceivable that another enzyme could target PR-Ub directly?

12. The argument for using Ub-VME to characterize low affinity ubiquitin-binding sites would be better supported by the example of Parkin in Wauer et al, 2015, rather than the self-citations of DUB structures.

13. Details of any cryoprotectants used for the crystallographic studies were left out of the methods section.

14. Ligand omit maps densities should be shown to support the ADPR-bound structures, as well as the five water molecules bound into the MavL active site that are discussed in Fig 5.

Reviewer #3 (Remarks to the Author):

The authors have identified MavL to bind ubiquitin and to remove ADP-ribosyl moiety from mono-ADP-ribosylated ubiquitin at its Arg-42 residue. They crystallized MavL in complex with ADP-ribose to reveal the structural basis for the arginine-specific de-ADP-ribosylation. They determined that MavL belongs to DUF4804 class with orthologues in some eukaryotes including *Drosophila*. Crystal structures of two *Drosophila* orthologues support the arginine-specific de-ADP-ribosylation by the members of DUF4804 class, an apparently different class among macrodomain-containing proteins.

The authors' findings fill the cap in the noncanonical ubiquitination via ADP-ribosylation by the action of SidE family effectors in that MavL is a missing linkage to reverse ubiquitin ADP-ribosylation, the first step in the phosphoribosyl (PR)-ubiquitination. Therefore, the manuscript in principle seems to advance our understanding of the PR-ubiquitination significantly both in fundamental biology and in pathogenesis associated with it. Despite the potential merits by this manuscript, some points need to be clarified.

MAJOR POINTS

Figure 1A: It would be desirable if the authors provide the entire data set by mass spectrometry analysis as a supplementary data. Also desirable is that the authors specify whether their proteomic system was able to capture the known ubiquitin interactors in *L. pneumophila*. Was MavL one of the top matches from the proteomic analysis? If MavL was not the top novel interactor, what criteria did the authors employ to select MavL? The figure legend should specify that orange circle, blue circle and cyan pentagon represent proteins in *L. pneumophila* proteome.

Figure 2: Could the authors provide data for the statement that they could not detect any ADP-ribosylation on UBE2Q1, at least as a part of the Supplementary Information?

Figure 3: Could the authors investigate the activity of MavL on the ADP-ribosylated ATP/ADP translocases? Larg1 and MavL are claimed to belong to the same broad arginine-specific macrodomain enzyme class but apparently show differential substrate specificities at protein level.

Figure 4 and Discussion (3rd paragraph on page 9): The authors stated that crystallization of MavL-UbVME yielded an apparently inactive structure presumably due to crystal contact artifacts. Could the authors attempt to use small-angle X-ray scattering (SAXS) to derive a structural model in solution for the MavL-UbVME? Such a model would tell the authors whether MavL-UbVME could assume an active-like state in solution.

Figure 6: What about comparison of MavL with ALC1-like proteins? As the authors stated in the Introduction, TARG1 and DarG in the ALC1-like class cleave N-glycosidic linkage in DNA substrates. A (brief) statement on the structural and sequence motif comparison would be desired.

Discussion: The authors claim that Larg1, a macrodomain-containing metaeffector also catalyzing arginine de-ADP-ribosylation of mitochondrial ADP/ATP translocases, is related to MavL. Given the observations that both Larg1 and MavL catalyze arginine de-ADP-ribosylation but that Larg1 and MavL apparently bind different mono-ADP-ribosylated proteins, it would be interesting to see some speculation on how these two proteins might recognize different mono-ADP-ribosylated protein substrates from structural aspects. More in-depth discussion on how Larg1 and MavL would function on different ADP-ribosylated protein substrates from structural perspectives would benefit the scientists in the related fields.

MINOR POINTS:

Title: Should the word "conversed" have been "conserved"?

Page 4, line 3: The letter "N" in the "N-glycosidic bond" needs to be italicized, consistent with standard chemical nomenclature.

Page 11, Section "Complex formation, crystallization, and data processing": Please specify the numbers for "equal molar amounts of UbVME and MavL42-435".

Page 15, Section "Quantification and statistics analysis": Please specify whether "triplicates" are technical or biological ones.

SUGGESTIONS:

Name of the DUP4804 class of macrodomain-enzymes: it might be more informative if the authors call the DUP4804 class as "macroR" class, based on the findings the authors reported in this manuscript.

Organization of the manuscript: It may not be essential for the authors to mention MavC/MvcA in

the Introduction.

Reading flow of the manuscript: The reviewer noticed many sentences are too long with many phrases in a sentence. The authors are encouraged to use shorter sentences to facilitate the reading flow of the manuscript.

Reviewer #1 (Remarks to the Author):

In the manuscript entitled “Legionella metaeffector MavL reverses ubiquitin ADP-ribosylation via a conserved arginine-specific macrodomain”, the authors clearly showed that MavL harbors (ADP-ribosyl)hydrolase activity having substrate preference to ADP-ribosylated ubiquitin, formed by mART activity of SidE family Legionella effectors. They also showed that MavL has metaeffector function against the SidE family effectors but the experimental validation was limited in HEK293T cells producing MavL by transient transfection. Structural analysis of ADPR-bound MavL identified residues forming the ADPR-binding pocket, while that of MavL-Ub is not great because good co-crystals were not available. Moreover, the authors analyzed CG2909 and CG3568 from *Drosophila*, both of which as well as MavL belong to DUF4804 family, and the authors concluded that the DUF4804 family represents a new class of Arg-specific macrodomains harboring (ADP-ribosyl)hydrolase activity.

Major comments

1. (Figure 2 panel BC) To demonstrate that MavL functions as a metaeffector against the SidE family, the authors ectopically expressed MavL or its catalytically-dead derivative in cells. To support the important conclusion of this paper, the experiment should be carried out in more physiological systems, such as infection using MavL catalytically-dead (or KO) and/or MavL overproducing strains.

Thanks for the comment. We performed an infection experiment to better access the function of MavL under a more physiological setting. We found that infection with $\Delta mavL$ or MavL^{D333A}-expressing Legionella strains results in higher level of ADPR-Ub in HEK 293T cells (Figure 2B), whereas the WT or $\Delta mavL$ (pMavL) strains results in lower level of cellular ADPR-Ub in the same infected cell culture system. These results align with our biochemical and other biological experiments, and we believe this new piece of data provides further support of MavL's role as a metaeffector (Figure 2B).

2. The authors should provide the data indicating that (over)production of mART-dead SdeA does not result in yeast toxicity as a control experiment.

Thank you for this feedback. We would like to point out that a previous study has shown that mART-dead SdeA does not result in yeast toxicity (please see reference 3). On the other hand, we also recognize that the same assay in these papers were performed with serial dilution plating and western blotting to show expression (as reviewer two pointed out, major concern 5), therefore we repeated our assay in the same fashion for better comparison (Figure 2E), in which we observed similar results showing that MavL, but not its catalytically impaired mutant, can rescue the toxicity in yeast caused by SdeA (or mART-dead SdeA) expression.

Minor comments

3. The entire results of the proteome-wide Ub-interactor screening (Figure 1 panel A) should be presented as a supplemental table, as the data would be good resources.

We certainly appreciate this comment. We have now deposited the proteomics datasets (as .raw files) in Mass Spectrometry Interactive Virtual Environment (MassIVE) database (identifier: MSV000091781). Additionally, we provided the Table S1 listing the Ub-interacting *Legionella* effectors identified from our proteomics experiments.

4. Description of full-length MavL and MavL42-435 is confusing especially in figures. For example, “MavL”'s in figure 1 panels B-D should be labeled as “MavL42-435”.

Thank you for pointing this out. We have carefully gone through our manuscript and ensured that all the figure labels are consistent and correct.

5. (Figure 1 panel A) what do the red texts (+HA-Ub, +buffer) mean?

These two samples were controls for our proteomics experiments. The sample of HA-Ub incubated with *Legionella* lysate identifies *Legionella* proteins that are not covalently captured by the probes, whereas the buffer incubated with *Legionella* lysate identifies *Legionella* proteins that are non-specifically enriched.

6. (Figure 2 panel F) Although I agree that this is a piece of important data which should be published, this data is nothing to do with the main logic of this figure/paper. Supplemental figure is a proper place to have this panel.

We appreciate this concern and understand that this piece of data may cause confusion in its current place. We agree that this should be moved to SI, and it is currently in SI Figure S3A.

7. (Figure 3 panel A) In Coomassie gels bands of MavL42-435 and ARH1 are readily visible, but not Larg1. It is nice to add a (native or denature) gel image showing that comparable amounts of Larg1 is included in the reactions.

Thanks for the comment. Larg1 does not travel through the native PAGE gel in our setting, therefore we updated an SDS-PAGE gel to show the amount of enzyme loaded (Figure 3A). Additionally, we included the loading of CG2909 and CG3568 for this assay as well (see Figure S6A).

8. (Figure 7 panel D) To achieve proper sequence alignment is one of the most important prerequisites of phylogenetic analysis. I believe that sensible alignments containing remote Larg1 sequence is pretty much hard or even impossible to obtain. How do the authors align sequences and how the alignment is validated? Furthermore, the authors do not provide any details about the phylogenetic analysis. Which evolution model is employed to confer the ML tree and why? No statistical validation of the tree such as bootstrap supports is indicated. The current tree is nonsense and should be removed from the figure.

Thanks for the feedback. We agree that the current phylogenetic analysis lacks rigorous validation and therefore we removed this from our manuscript to avoid confusion. We believe its removal does not alter the manuscript in any appreciable way.

9. Citations to ref 9 are missing in many places regarding Larg1 in texts.

We updated our manuscript accordingly and added ref 9 in the missing places.

Reviewer #2 (Remarks to the Author):

In their manuscript “Legionella metaeffector MavL reverses ubiquitin ADP-ribosylation via a conserved arginine-specific macrodomain”, Zhang et al. discover and characterize a new class of ADP-ribosylhydrolase enzymes unrelated to those currently known in humans. Legionella pneumophila secretes numerous effector proteins upon infection, some of which have been recently characterized to produce new ubiquitin modifications and linkages. The SidE family of Legionella E3 ligases ubiquitinate substrate proteins using an R42 ADP-ribosylated ubiquitin intermediate that is then conjugated onto target serineresidues via a phospho-ribose linkage. The Legionella deubiquitinases DupA and DupB reverse SidE-mediated ubiquitination by removing PR-Ub from substrates. Using a proteomics workflow, Zhang et al. identify the Legionella effector MavL as a ubiquitin-interacting protein, and provide evidence that it restores ADPR-Ub to its native form. They go on to suggest that MavL acts as a metaeffector, controlling the activity of the SidE ligase family. Following extensive structural characterization, the authors reveal that the arginine-specific macrodomain of MavL represents a novel class in this enzyme family, with structurally-related examples observed in eukaryotes including Drosophila melanogaster. This manuscript is mostly well written and the figures are easy to follow. It will be of interest to microbiologists and structural biologists across both the ubiquitin and ADPR fields. While the work is of high interest and quality, there are several issues that should be resolved. Primary among them are MavL’s role as a metaeffector and its specificity toward ubiquitin. This manuscript could be a candidate for publication in Nature Communications if the following concerns are addressed:

Major Concerns:

1. The first figure describes a mass spectrometry-based approach to finding Ub-interacting proteins in Legionella, but there is no indication in the manuscript of where the data will be deposited. These data should be made publicly available following publication.

We appreciate this feedback. Please refer to our response to [reviewer 1, minor concern 1] regarding this concern.

2. The crystal structure of MavL-UbVME does not accurately represent the enzyme:substrate complex, but the authors have discussed and justified findings as if it does. Based on their data, it is possible that the ubiquitin-bound structure is an artifact of crystallization. It is unclear what is learned from the ubiquitin-bound structure that could not be gleaned from the apo and ADPR-bound structures, and thus the authors should consider removing it from the manuscript entirely if it is indeed an artifact.

- a. R42 of ubiquitin is positioned far away from the MavL active site, and furthermore almost none of the mutations tested at the ubiquitin interface have an effect on MavL activity. These signs point toward the structure representing a crystallization artifact, which in this reviewer’s opinion should not be published as it will lead to future confusion in the field. Alternatively, the visualized ubiquitin-binding interface could be genuine, but distinct from the binding site for hydrolase activity. Have the authors tested any of their MavL mutants for a defect in ubiquitin binding?

We appreciate this comment. Indeed, we did not observe clear activity loss when mutating most of the interface residues in MavL. However, we would still like to keep the discussion of this structure for two reasons: 1) The MavL R370A mutant used for capturing ADPR-bound MavL was inspired by the ADPR-soaked MavL-UbVME structure. 2) The catalytically important F105 was found in this interface. Even though we do not understand the exact role of this residue played in MavL's activity, we think this structure logically led to the mutation of this residue. 3) We agree that at this point this structure looks like a crystallographic artifact, we cannot rule out the possibility that this may represent a certain pre- or post- catalytic state of the enzyme. Thus, we think that this structure may provide additional information if future study reveals more structural insights of MavL's catalysis.

On the other hand, to avoid the confusion created by putting this structure as a main figure, we moved the overall structure figure in SI (Figure S4), leaving only the mutational study figure as a main figure. We hope this rearrangement of figures avoid future concerns and confusions.

3. Several aspects of the authors' proposed enzymatic mechanism are overstated and should either be scaled back or solidified with more supporting data.

a. The authors show that both F105 and E107 are crucial for the removal of ADPR from ubiquitin, but the argument that they position the R42 sidechain (aliphatic and charged interactions, respectively) is speculative. Structural modeling of an Arg residue coming off the ADPR would indicate side chain direction and provide stronger support for the authors' statements. Additionally, mutation of these sites indicates their importance for activity, but not necessarily specificity. Have the authors determined if the F105 or E107 mutants are suddenly active toward other forms of ADP ribosylation?

Thanks for this feedback. We performed docking using MavL and Arg-ADPR and found that the program placed the arginine side chain in a way where it is coordinated by E107, D323, and T331, with F105 interacting with the aliphatic portion (Figure 4D). This result aligned with our statement regarding potential arginine binding pose, therefore we believe that F105 and E107 indeed helps positioning arginine. At this point, we believe that the specificity towards arginine MARYlation could be attributed to multiple structural features (for example, the depth of the active site), and the F105 and E107 mutants may not provide enough insights explaining the specificity.

b. The authors discuss the ADPR-bound structures as if they are bound to substrate, but in fact they are bound to product. This leads them to claim that they observe a catalytic water molecule in their ADPR-bound MavL structure (w4), but there is no biochemical support for this.

Thanks for this feedback. In our crystallization setup, the ADPR solution sample is a 50:50 mixture of α -ADPR and β -ADPR due to epimerization. MavL-ADPR complex has selected the α -anomer. Since the substrate ADPR-Ub has the same anomeric configuration, as has been observed in the crystal structure of ADPR-Ub in complex with SdeD (Reference 45, PDB: 6B7O), we propose that the APDR binding mode reflects interaction with the substrate at this anomeric center. We have now clarified this in the text (please see page xx). A similar example was seen for the MacroD-type (ADP-ribosyl)hydrolase.

i. The T331 sidechain is shown to coordinate w4 in Fig 5C/D but makes no contact to the ADPR molecule. If w4 were the catalytic water, why does substitution at T331 not effect MavL activity determined in Fig 4F?

We revisited the assay and optimized the amount of enzyme used. We found that T331A mutant cannot effectively catalyze the de-ADP-ribosylation at 100 nM, suggesting that this residue indeed plays a role in MavL activity. As the w4 coordinates to di-phosphates and the backbone of MavL, we believe that could be the reason why T331 plays less important roles in the catalysis compared to D333, for example.

ii. Are these waters also observed in similar positions in the CG2909 and CG3568 ADPR-bound structures?

Yes, we provided the figures showing these water molecules in Figure 5G and 5H.

c. Fig 5D shows D323 coordinating the R42/ribose connection, yet the authors also propose that E107 serves this function. Which is it?

We propose that E107 and D323 coordinate two separate amine groups in Arg. Our docking result (Figure 4D) also supports this hypothesis. We agree that this point was not clearly stated in the original manuscript, therefore, we provided an additional figure presenting this proposed arginine coordination (Figure 4E)

d. How would the authors explain the lack of an effect from removing the ability of H142 and Y265 to stack with the adenosine ring of ADPR?

We did not observe any loss of activity with H142A/Y265A mutant even using low concentration of MavL. As these two residues were not involved in distal ribose interaction, we believe that these two residues serve solely binding roles but not catalytic roles, and their mutation can be tolerated. This suggests adenine binding does not appreciably contribute to substrate affinity.

4. Regarding MavL's function as a metaeffector, the authors show that cells ectopically expressing MavL are defective for SdeA-mediated ubiquitination following either Legionella infection or SdeA ectopic expression. While consistent with this possibility, both of these experiments are far from native conditions. The authors already have a Δ mavL Legionella strain; following infection do you observe an increase in either ADPR-Ub, Ub-Rab33b, or total PR-ubiquitination compared to WT Legionella? Regarding the interpretation of this assay, have you formally demonstrated that MavL cannot hydrolyze PR-ubiquitinated substrates?

We performed a de-ubiquitination assay using PR-ubiquitinated Rab33b (Figure 1E) and showed that MavL cannot cleave off ubiquitin from Ub-PR-Rab33b.

5. The yeast toxicity assay should be repeated with serial dilution plating and western blotting to show expression levels of the various constructs.

We repeated this assay with suggested format and updated the figure (Figure 2E).

6. The BLI curve fits for the MavL:ubiquitin interaction do not look very good. Can the authors explain this?

Thanks for this comment. We attribute this to the slow association of Ub and MavL. We also attempted re-analyzing the data with

7. There are several inconsistencies in the reported western blot experiments:

a. In Fig 3B, why does the addition of more MavL increase the amount of ADPR detected for PARylated His-PARP1? If the amount of MavL is the only thing changing, and MavL is not reactive to this species of ADPR-protein, shouldn't the lanes all look identical?

b. There appears to be a loading issue for the blots presented in Fig 3C. The 6His blot should be consistent but appears to diminish with increasing concentration of MavL.

c. There is inconsistent loading in several other control blots including S4B, S4C.

Thanks for pointing these out. We repeated these assays and updated the figures. We believe the updated figures showed clear loading of these assays and better support our conclusions.

8. As further support that MavL is specifically a ubiquitin-directed ADPR hydrolase, it would be useful to include ADPR-Ub as a substrate in the luminescence assay presented in Fig 6D for comparison to the ADPR-Arg substrate.

Thanks for the comments. We performed this assay and found that, interestingly, MavL does not exhibit clear activity differences hydrolyzing ADPR-Arg peptide and ADPR-Ub (Figure S6D). As the substrate recognition was mostly driven by the ADPR moiety (based on our binding affinity characterization), we think that ubiquitin binding may not contribute to MavL catalysis as significantly. This is further supported by the fact that MavL exhibited similar de-ADP-ribosylation activity against ADPR-ANT1 (an assay suggested by reviewer 3, major concern 3). On the other hand, we also think that in the context of protein substrates, local conformation around ADPR-Arg could play a role by "rejecting" the enzyme. This is supported by the low/no MavL activity processing ADPR-Gdp2h and ADPR-Rab5. Therefore, we revised the manuscript and made statements as follows: "It is worth noting that MavL does not exhibit significant differences hydrolyzing ADPR-Ub compared to ADPR-Arg peptide (Figure S6D) and exhibits similar activity towards ADPR-ANT1 compared to Larg1 (Figure 3B). This lack of stringent protein-level selectivity could be attributed to the higher binding affinity of MavL to ADPR than to Ub, which implies that the substrate recognition is likely dominated by ADPR. On the other hand, Larg1 hydrolyzing ADPR-Arg at a clearly faster rate than ADPR-Ub (Figure and S6E), indicating the possibility that the local structure near R42 of Ub might dismiss de-ADP-ribosylating enzymes other than MavL. The fact that MavL exhibited weak activity towards MARYlated Gdp2h but not ADP-ribosylated Rab5 also supports the speculation that this macrodomain class cares for protein substrates beyond just MARYlated arginine. It appears that there is a balance between the recognition of MARYlated arginine and the accommodation of local structure near the ADP-ribosylation, which can possibly be achieved by the variable regions and the insertions of this enzyme class. Further studies aimed at co-crystallization of substrates with the enzyme will help elucidate this point."

Minor Concerns:

1. In the majority of structure figures, protein subunits are not labeled and coloring is not indicated. This would clarify things and orient the reader better.

a. Additionally, the coloring used to highlight interactions in the CG3568 structure presented in Fig. 7E/F are difficult to discern. The authors should consider color choice for the structure and change the appropriate figures.

Thanks for the comment. We updated our figures with protein subunits labeled. The interactions in CG3568 structure were presented in a different color.

2. The authors should read the manuscript carefully before re-submission and remove extra words, check for grammar, correct nomenclature inconsistencies etc.

Thanks for the comments. We carefully went through the manuscript again and we believed that we have corrected the errors.

3. The methods section should include a description of how the native PAGE was performed.

We added a detailed description regarding how the native PAGE was run.

4. Authors show an interaction between UBE2Q1 and MavL but do not explore this further. This seems out of place in this manuscript.

Thank you for the comment. We decided to move this piece of data to SI, together with the data showing that we did not detect ADP-ribosylation on UBE2Q1. These data are shown in Figure S3B and S3C.

5. It is not clear from the figure legends that “V” denotes empty vector control.

We added a description in the figure legends to clarify this.

6. The orientation in Fig 5C shows the A313 backbone amide coordinating w4, but the authors show the backbone carbonyl coordinating w4 in Fig 5D.

Thanks for pointing this out. w4 is coordinated to the backbone carbonyl. We understand that the current view of Figure 5C causes some confusion, and we updated this figure by providing an alternative view.

7. Figures 7D and 7E are presented out of order.

We decided to remove Figure 7D from this manuscript, due to lack of validation of our phylogenetic analysis (see review 1, minor concern 8).

8. Labeling the two loops involved in ADPR binding in the topology diagrams of Fig 7A would be helpful.

Thanks for the comments. We updated the topology diagrams to clearly show the positions of di-phosphate-binding loop and catalytic loop.

9. References to Table S1 should be included in the areas of text that describe each crystal structure.

Thanks for the comment. We updated our manuscript and added these references to Table S1 at these places.

10. On page 8, there is a description of the conservation of E107 that references Fig 7F. I believe this should be referencing Fig 7E.

Thanks for pointing this out. We made corrections accordingly.

11. In the discussion, it is unclear why this work suggests the existence of an additional enzyme that converts PR-Ub to ADPR-Ub. Is it not conceivable that another enzyme could target PR-Ub directly?

Thanks for pointing this out. We revised our manuscript with the following statement: “At this point, it is still not clear what would be the metabolic fate of PR-Ub. It is possible that an enzyme could directly hydrolyze the phosphoribose moiety from PR-Ub, but the discovery of MavL suggests an alternative pathway regenerating Ub from PR-Ub, where an unknown enzyme would convert PR-Ub into ADPR-Ub for de-ADP-ribosylation by MavL.”

12. The argument for using Ub-VME to characterize low affinity ubiquitin-binding sites would be better supported by the example of Parkin in Wauer et al, 2015, rather than the self-citations of DUB structures.

Thanks for the comment. We revised our manuscript with the appropriate citation.

13. Details of any cryoprotectants used for the crystallographic studies were left out of the methods section.

The crystals were directly looped from the drop and flash-frozen without cryoprotectant. We added descriptions to clearly state this point.

14. Ligand omit maps densities should be shown to support the ADPR-bound structures, as well as the five water molecules bound into the MavL active site that are discussed in Fig 5.

Thanks for this comment. We updated our figures with corresponding omit maps (Figure 4A, 4C, 5G and 5H).

Reviewer #3 (Remarks to the Author):

The authors have identified MavL to bind ubiquitin and to remove ADP-ribosyl moiety from mono-ADP-ribosylated ubiquitin at its Arg-42 residue. They crystallized MavL in complex with ADP-ribose to reveal the structural basis for the arginine-specific de-ADP-ribosylation. They determined that MavL belongs to DUF4804 class with orthologues in some eukaryotes including *Drosophila*. Crystal structures of two *Drosophila* orthologues support the arginine-specific de-ADP-ribosylation by the members of DUF4804 class, an apparently different class among macrodomain-containing proteins.

The authors' findings fill the gap in the noncanonical ubiquitination via ADP-ribosylation by the action of SidE family effectors in that MavL is a missing linkage to reverse ubiquitin ADP-ribosylation, the first step in the phosphoribosyl (PR)-ubiquitination. Therefore, the manuscript in principle seems to advance our understanding of the PR-ubiquitination significantly both in fundamental biology and in pathogenesis associated with it. Despite the potential merits by this manuscript, some points need to be clarified.

MAJOR POINTS

Figure 1A: It would be desirable if the authors provide the entire data set by mass spectrometry analysis as a supplementary data. Also desirable is that the authors specify whether their proteomic system was able to capture the known ubiquitin interactors in *L. pneumophila*. Was MavL one of the top matches from the proteomic analysis? If MavL was not the top novel interactor, what criteria did the authors employ to select MavL? The figure legend should specify that orange circle, blue circle and cyan pentagon represent proteins in *L. pneumophila* proteome.

We appreciate this feedback. Please refer to our response to [reviewer 1, minor concern 1] regarding this concern.

Figure 2: Could the authors provide data for the statement that they could not detect any ADP-ribosylation on UBE2Q1, at least as a part of the Supplementary Information?

Thanks for the comments. We included our data regarding UBE2Q1 in the SI (see Figure S3), together with the MavL:UBE2Q1 immunoprecipitation. We showed that ectopically expressed UBE2Q1 in HEK 293T cells does not harbor ADP-ribosylation. Additionally, upon co-incubation with either *L. pneumophila* lysate or HEK 293T cell lysate, UBE2Q1 does not show ADP-ribosylation.

Figure 3: Could the authors investigate the activity of MavL on the ADP-ribosylated ATP/ADP translocases? Larg1 and MavL are claimed to belong to the same broad arginine-specific macrodomain enzyme class but apparently show differential substrate specificities at protein level.

Thanks for the comments. We performed this experiment and found that, interestingly, Larg1, MavL, and ARH1 exhibit similar de-ADP-ribosylation activity towards immunoprecipitated ADPR-ANT1. However, we also want to point out that ANT1 locates on mitochondrial inner membrane, with ADPR-Arg facing the mitochondrial matrix. A previous study [ref] showed that ARH1 cannot remove ADP-ribosylation on ANT1 when mitochondria was used as the substrate, indicating that the ability of Larg1 to enter mitochondria plays a role in this de-ADP-ribosylation. As MavL was found to localize in cytoplasm (reference 9), we believe that this de-ADP-ribosylation is unlikely to happen in cells.

Figure 4 and Discussion (3rd paragraph on page 9): The authors stated that crystallization of MavL-UbVME yielded an apparently inactive structure presumably due to crystal contact artifacts. Could the authors attempt to use small-angle X-ray scattering (SAXS) to derive a

structural model in solution for the MavL-UbVME? Such a model would tell the authors whether MavL-UbVME could assume an active-like state in solution.

Thanks for this suggestion. We performed SEC-SAXS (Figure S5) and found that MavL-UbVME is in monomeric state in solution. We also attempted reconstructing the 3D electron density map of MavL-UbVME, yet we did not find this covalent complex mimicking an active-like state in solution. Additionally, we found that MavL-UbVME exhibited similar activity compared to MavL (Figure S4G), further supporting that it is unlikely to represent an active-like state. As reviewer 2 suggested, we have moved our statements regarding MavL-UbVME to SI (Figure S4 and S5) to avoid confusion.

Figure 6: What about comparison of MavL with ALC1-like proteins? As the authors stated in the Introduction, TARG1 and DarG in the ALC1-like class cleave N-glycosidic linkage in DNA substrates. A (brief) statement on the structural and sequence motif comparison would be desired.

Thanks for this comment. As there is no reported sequence motif for ALC1-like class, we compared the ADPR-binding sites of MavL, TARG1, and DarG. We provided an active site comparison showing that MavL does not have the catalytic lysine residues found in TARG1 and DarG (Figure 5A). In the manuscript, we added the following statement: “As mentioned before, ALC1-like class employs a lysine-dependent de-ADP-ribosylation mechanism involving a ring-opening Schiff base intermediate. Given no conserved sequence motif has been reported for this class, we compared ADPR-bound structures of MavL, TARG1 (PDB: 4J5S), and DarG (PDB: 5M3E). Superposition of ADPR moieties in these structures clearly showed the catalytic lysine residues in TARG1 (K84) and DarG (K80), whereas MavL has an aspartate residue (D315) at the same position (Figure 5A). Additionally, the catalytic loop of MavL does not contain any lysine residue (Figure 4A), further suggesting that MavL adopts a catalytic mechanism distinct from the ALC1-like class.”

Discussion: The authors claim that Larg1, a macrodomain-containing metaeffector also catalyzing arginine de-ADP-ribosylation of mitochondrial ADP/ATP translocases, is related to MavL. Given the observations that both Larg1 and MavL catalyze arginine de-ADP-ribosylation but that Larg1 and MavL apparently bind different mono-ADP-ribosylated proteins, it would be interesting to see some speculation on how these two proteins might recognize different mono-ADP-ribosylated protein substrates from structural aspects. More in-depth discussion on how Larg1 and MavL would function on different ADP-ribosylated protein substrates from structural perspectives would benefit the scientists in the related fields.

Thanks for this comment. During the revision, we found that MavL can process ADPR-ANT1 as efficiently as Larg1. Additionally, we also found there is no significant activity difference of MavL processing ADPR-Ub or ADPR-Arg peptide. These results showed that the specificity of MavL may not be as stringent. Therefore, in discussion, we provided statement as follows: “To date, most characterizations of the substrate specificities of MARYlation removing macrodomains relied on the residues that are ADP-ribosylated, posing an intriguing question on whether macrodomains have preferences at the protein level. Our activity assays using synthetic

MARylated substrate panel showed a clear preference of this new macrodomain class towards hydrolytic reversal of arginine MARylation, with comparable activity among these four enzymes regarding processing the MARylated arginine peptide (Figure 5D). However, MavL interacts with Ub and effectively removes ADPR from ADPR-Ub, whereas Larg1, CG2909 and CG3568 exhibited rather weak activity hydrolyzing ADPR-Ub (Figure 3A and S6A). It is worth noting that MavL does not exhibit significant differences hydrolyzing ADPR-Ub compared to ADPR-Arg peptide (Figure S6D) and exhibits similar activity towards ADPR-ANT1 compared to Larg1 (Figure 3B). This lack of stringent protein-level selectivity could be attributed to the higher binding affinity of MavL to ADPR than to Ub, which implies that the substrate recognition is likely dominated by ADPR. On the other hand, Larg1 hydrolyzing ADPR-Arg at a clearly faster rate than ADPR-Ub (Figure and S6E), indicating the possibility that the local structure near R42 of Ub might dismiss de-ADP-ribosylating enzymes other than MavL. The fact that MavL exhibited weak activity towards MARylated Gdp2h but not ADP-ribosylated Rab5 also supports the speculation that this macrodomain class cares for protein substrates beyond just MARylated arginine. It appears that there is a balance between the recognition of MARylated arginine and the accommodation of local structure near the ADP-ribosylation, which can possibly be achieved by the variable regions and the insertions of this enzyme class. Future investigation on the biological substrates for CG2909 and CG3568 would provide deeper insights into the substrate specificity and recognition of this type of macrodomains.”

MINOR POINTS:

Title: Should the word "conversed" have been "conserved"?

Thanks for pointing this out! This is indeed a typo.

Page 4, line 3: The letter "N" in the "N-glycosidic bond" needs to be italicized, consistent with standard chemical nomenclature.

Thanks for the comment, we have carefully gone through the manuscript and fixed these errors.

Page 11, Section "Complex formation, crystallization, and data processing": Please specify the numbers for "equal molar amounts of UbVME and MavL42-435".

We added description stating that MavL and UbVME were incubated at a 1:1 molar ratio.

Page 15, Section "Quantification and statistics analysis": Please specify whether "triplicates" are technical or biological ones.

We specified these triplicates as biological triplicates.

SUGGESTIONS:

Name of the DUP4804 class of macrodomain-enzymes: it might be more informative if the authors call theDUP4804 class as "macroR" class, based on the findings the authors reported in this manuscript.

We appreciate this suggestion. In our paper, only four enzymes (Larg1, MavL, CG2909, and CG3568) in this class have been structurally and biochemically characterized. Even though we

provided evidence that this protein class is likely to harbor Arg-specificity, we believe some future studies are needed to further solidify our findings. Thus, we would like to keep the name as it is.

Organization of the manuscript: It may not be essential for the authors to mention MavC/MvcA in the Introduction.

Thanks for the suggestion. We deleted this portion in the introduction to make it more clear.

Reading flow of the manuscript: The reviewer noticed many sentences are too long with many phrases in a sentence. The authors are encouraged to use shorter sentences to facilitate the reading flow of the manuscript.

Thanks for the comment. We went through the manuscript and broke down some long sentences. We think the current manuscript should be easier to read.

Reviewer #2 (Remarks to the Author):

The authors have done a nice job of further probing the role of MavL in Legionella biology and in determining its enzymatic properties. The primary issue that is left standing, however, is to illustrate the specificity (if there is one) toward ubiquitin. The question remains whether MavL is an ADPR hydrolase that happens to counter the activity of SidE, or whether this process is truly specific to ubiquitin modifications. Along these lines, the authors did not address a comment in the first review surrounding the effect of chosen MavL mutations on the interaction with ubiquitin. If ubiquitin binding can be disrupted, this opens doors to new experiments (perhaps in the yeast co-expression system) that demonstrate the importance of ubiquitin specificity. Furthermore, the authors did not fully respond to the query regarding the poor fits of BLI binding data that describe the MavL:ubiquitin interaction, which are arguably the main link between these two proteins aside from reactivity with the VME probe. Addressing these points seems essential for the conclusion that MavL is a ubiquitin ADPR hydrolase, as indicated in the title of the manuscript.

Minor comments:

- 1) A structural view similar to Fig. 4A, with an overlay from the ubiquitin-bound structure, would be helpful for understanding the relationships between these binding sites.
- 2) The UBE2Q1 data still feels very out of place with the flow of the manuscript.
- 3) The new western blot presented in Figure 2E needs labels above or below the lanes.
- 4) The authors alternate between nomenclature for (ADP-ribosyl)hydrolase and de-ADP-ribosylation. Consistency on this matter would help the reader.
- 5) The figure references on page 6 for Fig. S4 presenting the ubiquitin-bound MavL structure are incorrect.

Reviewer #3 (Remarks to the Author):

In this revised manuscript, the authors added new data and answered the issues raised by the reviewer to some extent. However, there are some remaining points.

Table S1: No details are provided. For instance, no explanation for gene names colored in red; for abbreviations in the table such as Ub-ABP, Prg, VME and VS even though they are mentioned in the text. A title row is missing on the top line of the table: "Buffer, Ub, Prg, VME, VS". The gene name "Lpg2526" should be followed by "(MavL)". The sentence "Numbers indicate ..." should not appear as a part of the title; it should be relocated at the bottom of the table with appropriate superscript symbols in the table. It should be described that the entire dataset has been deposited to MassIVE with the identifier MSV000091781 at the bottom of the table.

Figure 3: It seems that the authors added a new panel (B) in the revised manuscript, but they did not specify it in the rebuttal letter, rendering it difficult for the reviewer to unequivocally ensure which part of the figure the authors discuss. What is specifically "[ref]" in the rebuttal letter?

Figure 4A: ADP-ribose (apparently shown as stick models) should be indicated in the legend. K80 of DarG and K84 of TARG1 apparently point differently from that of D315 of MavL. Can the authors mention about this? What about other active site residues from multiple sequence alignment? The substrate interaction map in Figure 4E reveals that D315 is unlikely to be a primary residue in the active site. Current simple description "...further suggesting that MavL adopts a catalytic mechanism distinct from the ALC1-like class" does not seem to be appropriate and convincing.

Figure S5: How was the overall chromatogram from size exclusion chromatography? The authors should show the goodness of fit (χ^2) for rigid body modelling of the "inactive" structural model in the 3D electron density or molecular envelope derived from the SEC-SAXS data to support their claim that MavL-UbVME is unlikely to represent an active like-state. Calculated curve from the fitted model should be shown on top of the raw data in panel (A). To obtain structural insights on ADPR-Ub:MavL complex, the authors could attempt to prepare a catalytically dead mutant of MavL

and measure SEC-SAXS data on the ADPR-Ub:MavL complex. In that way, the authors can provide more in-depth sights on substrate preference/selectivity issue discussed in the revised manuscript.

Discussion: The paragraph discussed potential preference of MavL beyond ADPR moiety vaguely. More in-depth statements would be desired with citing similar examples for the "beyond just residue-level specificity". Considering that MavL interacts with Ub and removes ADPR from ADPR-Ub, the authors should discuss further on how the interaction of MavL with Ub might contribute to the potential substrate preference from structural and/or biochemical perspectives.

Reviewer #5 (Remarks to the Author):

I was asked to comment on the proteomics aspects of this manuscript after the initial round of review. I am not suited to comment on the biology of this system or this study, so I will leave that to the other reviewers. It is good the authors uploaded the data to the public repository MassIVE. However, the new supplementary table included is insufficient. First what do "MS counts" mean? Is that MS/MS spectra identified? The authors should upload the raw results and processed tables from their proteomics experiments as supplemental Excel file, not a table in a pdf. Those raw results and processed results should also be uploaded to MassIVE, and reviewer credentials to login and inspect the files should be provided.

Reviewer #6 (Remarks to the Author):

The authors have addressed the concerns of the first reviewer. However, I trust the authors, the new Figure 2B is just a single experiment and is not statistically quantitative to support MavL reduced the amount of ADPR-Ub.

Reviewer #2 (Remarks to the Author):

The authors have done a nice job of further probing the role of MavL in Legionella biology and in determining its enzymatic properties. The primary issue that is left standing, however, is to illustrate the specificity (if there is one) toward ubiquitin. The question remains whether MavL is an ADPR hydrolase that happens to counter the activity of SidE, or whether this process is truly specific to ubiquitin modifications. Along these lines, the authors did not address a comment in the first review surrounding the effect of chosen MavL mutations on the interaction with ubiquitin. If ubiquitin binding can be disrupted, this opens doors to new experiments (perhaps in the yeast co-expression system) that demonstrate the importance of ubiquitin specificity. Furthermore, the authors did not fully respond to the query regarding the poor fits of BLI binding data that describe the MavL:ubiquitin interaction, which are arguably the main link between these two proteins aside from reactivity with the VME probe. Addressing these points seems essential for the conclusion that MavL is a ubiquitin ADPR hydrolase, as indicated in the title of the manuscript.

Thank you for the positive assessment and we appreciate the comments regarding demonstration of specificity. Since we were unsuccessful in capturing non-covalently bound ubiquitin to MavL through co-crystallization, we are unable to identify MavL mutants that would show impaired ubiquitin binding. However, we thought we could approach the specificity issue through substrate mutagenesis. If ubiquitin mutants could be found that show impaired binding to the enzyme, we could probe the specificity question by looking at ADP-ribosylated derivatives of these mutants in MavL-catalyzed (ADP-ribosyl)hydrolysis. To this end, we generated five ubiquitin point mutants targeting residues proximal to the arginine (R42) carrying the ADPR group. Among these, we found that Ub^{Q40E} and Ub^{E51K} mutants were impaired in binding to MavL (Figure S2), as revealed in ITC experiments. Fortunately, one of these mutants (Ub^{Q40E}) retained the ability to be MARYlated at R42 by the SdeA^{mART}, allowing us to purify the corresponding ADPR-derivative. Indeed, this mutant (ADPR-Ub^{Q40E}) exhibited lower activity when treated with MavL, compared to WT ubiquitin and the mutants retaining binding to MavL (Figure S2F and Figure 3B). We have added these new results in the main text of the manuscript. We believe this new piece of data provides evidence that ubiquitin binding is necessary for optimal activity of MavL. These results further help support the notion that MavL catalysis is determined by both residue-level and protein-level selectivity, even though the interaction with the nucleotide portion contributes more to the substrate recognition.

Respectfully, we would like not to pursue yeast experiments, which could have been the correct course of action if we had found appropriate MavL mutants.

With regard to the BLI comments, we do not have a good handle on the fitting issue. It could be due to the modest binding between the two proteins. Unrelated to this work, we have made a number of BLI measurements on other interactions involving ubiquitin, such as with deubiquitinases. We see the same thing: when the interaction is weaker the fits tend to be poorer. To alleviate the concern of poor BLI fitting, we re-performed all our binding experiments using ITC, which showed a K_d of 88.3 μ M between MavL and ubiquitin, similar to the K_d provided in our previous manuscript (74 μ M). The ITC results agree with the BLI results and ITC also allowed us to look at ubiquitin mutants, as mentioned above. We could not repeat the BLI experiment in a timely manner due to instrumentation limitations, thus we removed this piece of data to avoid further confusion.

Minor comments:

1) A structural view similar to Fig. 4A, with an overlay from the ubiquitin-bound structure, would be helpful for understanding the relationships between these binding sites.

Thanks for the comment. We modified Figure S3E to show this view.

2) The UBE2Q1 data still feels very out of place with the flow of the manuscript.

Thanks for the feedback. We moved this section to discussion. We think this rearrangement should provide a better flow of the manuscript while keeping this piece of data.

3) The new western blot presented in Figure 2E needs labels above or below the lanes.

Thanks for the comments. We have added labels to the blots.

4) The authors alternate between nomenclature for (ADP-ribosyl)hydrolase and de-ADP-ribosylation. Consistency on this matter would help the reader.

Thanks for this suggestion. We have gone through the manuscript and made changes to keep consistent nomenclature.

5) The figure references on page 6 for Fig. S4 presenting the ubiquitin-bound MavL structure are incorrect.

We have gone through this section again and made changes to reference the correct figures.

Reviewer #3 (Remarks to the Author):

In this revised manuscript, the authors added new data and answered the issues raised by the reviewer to some extent. However, there are some remaining points.

Table S1: No details are provided. For instance, no explanation for gene names colored in red; for abbreviations in the table such as Ub-ABP, Prg, VME and VS even though they are mentioned in the text. A title row is missing on the top line of the table: "Buffer, Ub, Prg, VME, VS". The gene name "Lpg2526" should be followed by "(MavL)". The sentence "Numbers indicate ..." should not appear as a part of the title; it should be relocated at the bottom of the table with appropriate superscript symbols in the table. It should be described that the entire dataset has been deposited to MassIVE with the identifier MSV000091781 at the bottom of the table.

Thanks for this comment. We understand that the current presentation of the mass spec data raised some concerns (please also see our response to Reviewer #5). Taking all the comments together, we uploaded Table S1 as a separate Excel file and provided additional explanations. As we cannot modify the existing MassIVE deposition, we created a new deposition with the following credentials:

Username: MSV000093623_reviewer

Password: Lp02UbABPs

Figure 3: It seems that the authors added a new panel (B) in the revised manuscript, but they did not specify it in the rebuttal letter, rendering it difficult for the reviewer to unequivocally ensure which part of the figure the authors discuss. What is specifically "[ref]" in the rebuttal letter?

We appreciate this comment and our apologies for the typographical error. We rearranged the order of Figure 3 in this new version of the revised manuscript. Specifically, we added a new panel

(B) demonstrating the specificity of MavL towards Ub (together with Figure S2). Please see our response to Reviewer #2 on this matter. Regarding the original panel (B), we moved it as the new panel (G), in which we demonstrate MavL's activity towards other arginine MARYlation (together with Figure 3E and 3F). We hope this new arrangement provides a better flow for the manuscript.

Figure 4A: ADP-ribose (apparently shown as stick models) should be indicated in the legend. K80 of DarG and K84 of TARG1 apparently point differently from that of D315 of MavL. Can the authors mention about this? What about other active site residues from multiple sequence alignment? The substrate interaction map in Figure 4E reveals that D315 is unlikely to be a primary residue in the active site. Current simple description "...further suggesting that MavL adopts a catalytic mechanism distinct from the ALC1-like class" does not seem to be appropriate and convincing.

Thanks for the comment. We believe this comment refers to Figure 5A, not 4A. To address the concern here, we added a line in the text stating the different side chain orientations across these three enzymes. We understand that the previous view did not clearly show the overall terminal ribose interactions in all three enzymes, therefore we provided additional views of the active sites. We hope these changes help highlight the differences between MavL and the ALC1-like macrodomain class.

Figure S5: How was the overall chromatogram from size exclusion chromatography? The authors should show the goodness of fit (χ^2) for rigid body modelling of the "inactive" structural model in the 3D electron density or molecular envelope derived from the SEC-SAXS data to support their claim that MavL-UbVME is unlikely to represent an active like-state. Calculated curve from the fitted model should be shown on top of the raw data in panel (A). To obtain structural insights on ADPR-Ub:MavL complex, the authors could attempt to prepare a catalytically dead mutant of MavL and measure SEC-SAXS data on the ADPR-Ub:MavL complex. In that way, the authors can provide more in-depth sights on substrate preference/selectivity issue discussed in the revised manuscript.

Thanks for the comment. We provided the overall SEC chromatogram in Figure S5 and the goodness of fit (χ^2). We also included the calculated curve from the fitted model.

Regarding the ADPR-Ub:MavL complex (with inactive mutant), we found that these two components do not stick together in SEC, therefore we believe it could be difficult to obtain meaningful data. As an alternative approach, we access the substrate preference/selectivity via multiple ubiquitin mutants. Please see our previous response (and response to Reviewer #2).

Discussion: The paragraph discussed potential preference of MavL beyond ADPR moiety vaguely. More in-depth statements would be desired with citing similar examples for the "beyond just residue-level specificity". Considering that MavL interacts with Ub and removes ADPR from ADPR-Ub, the authors should discuss further on how the interaction of MavL with Ub might contribute to the potential substrate preference from structural and/or biochemical perspectives.

Thanks for the comment. Based on our new experiments on Ub mutants, we modified our discussion as follows:

"To date, most characterizations of the substrate specificities of MARYlation removing macrodomains relied on the residues that are ADP-ribosylated, posing an intriguing question on whether macrodomains have preferences at the protein level. Our activity assays using synthetic MARYlated substrate panel showed a clear preference of this new macrodomain class towards

hydrolytic reversal of arginine MARYlation, with comparable activity among these four enzymes (Figure 5E). These synthetic peptides represent unfolded structural moieties presenting the MARYlated residues to the catalytic action of the enzymes, essentially reporting on residue-level activity. However, MavL interacts with Ub and effectively removes ADPR from ADPR-Ub, whereas Larg1, CG2909 and CG3568 exhibited rather weak activity hydrolyzing ADPR-Ub (Figure 3A and S5). The requirement of protein-level recognition is further demonstrated by the activity difference of MavL towards the MARYlated Ub mutants (Figure 3B). Furthermore, the observation that MavL exhibited weak activity towards MARYlated Gdp2h but not ADP-ribosylated Rab5 (Figure 3E and 3F) also supports the speculation that this macrodomain class cares for protein substrates beyond just MARYlated arginine. These results collectively emphasize the context of protein in addition to the MARYlated arginine element and suggest a particular evolutionary logic in the construction of these macrodomains. Specifically, the arginine selectivity could have appeared first in the macrodomain scaffold followed by further decoration with substrate-specific elements to achieve specific targeting. For individual arginine-specific macrodomain enzymes, it appears that there is a balance between the recognition of MARYlated arginine and the accommodation of local structure near the ADP-ribosylation, which can possibly be achieved by the variable region and the insertion present in the structure (Figure 6A). Future investigation on the biological substrates for CG2909 and CG3568 would provide deeper insights into the substrate specificity and recognition of this type of macrodomains..”

In addition, we added a few lines highlighting the fact that the gene encoding MavL is in proximity to other genes encoding Ub-related effectors (in the first paragraph of discussion):

“At this point, it is still not clear what would be the metabolic fate of PR-Ub. However, it is worth noting that *Legionella* effectors targeting relevant pathways tend to occur adjacently in the genome ^{3,12}. In the case of MavL (lpg2526), five other Ub-related effectors have been characterized, including SdjA ^{15,17} (lpg2508, polyglutamylase inactivating SidE family mART, except for SdeA), DupB ^{13,14} (lpg2509, deubiquitinase removing PR-Ub), SdcA ⁵⁵ (lpg2510, E3 ligase), SidC ⁵⁵ (lpg2511, E3 ligase), and LotC ^{56,57} (lpg2529, deubiquitinase). It is therefore tempting to speculate that a gene in proximity to MavL may encode an effector directly hydrolyzing the phosphoribosyl moiety from PR-Ub, or converting PR-Ub into ADPR-Ub for (ADP-ribosyl)hydrolysis by MavL.”

Reviewer #5 (Remarks to the Author):

I was asked to comment on the proteomics aspects of this manuscript after the initial round of review. I am not suited to comment on the biology of this system or this study, so I will leave that to the other reviewers. It is good the authors uploaded the data to the public repository MassIVE. However, the new supplementary table included is insufficient. First what do “MS counts” mean? Is that MS/MS spectra identified? The authors should upload the raw results and processed tables from their proteomics experiments as supplemental Excel file, not a table in a pdf. Those raw results and processed results should also be uploaded to MassIVE, and reviewer credentials to login and inspect the files should be provided.

Thanks for the feedback. In this round of revision, we submitted Table S1 as a separate Excel file with all the protein groups identified included as a separate sheet. We specified that the numbers are MS/MS spectral counts in the table. Due to current maintenance of the database, we were unable to modify our existing dataset. We therefore created a new deposition with all the files uploaded. The credentials are as follows:

Username: MSV000093623_reviewer

Password: Lp02UbABPs

Reviewer #6 (Remarks to the Author):

The authors have addressed the concerns of the first reviewer. However, I trust the authors, the new Figure 2B is just a single experiment and is not statistically quantitative to support MavL reduced the amount of ADPR-Ub.

We appreciate this feedback. We had repeated this experiment in triplicates. We now provide an additional figure (Figure 2C) showing quantification of ADPR-Ub:Ub ratio to support our statement that MavL reduces ADPR-Ub level during infection.

Reviewer #2 (Remarks to the Author):

The authors have adequately addressed my concerns.

Reviewer #3 (Remarks to the Author):

The authors have addressed the issues raised by this reviewer satisfactorily overall. However, some issues remain. The reviewer finds that the revised manuscript needs grammatical and organizational touches.

Below are the remaining concerns based on the rebuttal letter and the revised manuscript. Detailed recommendations for manuscript organization are also noted at the end of the comments.

COMMENTS ON THE REMAINING CONCERNS:

Table S1: In the first tab of the EXCEL file provided, the authors are strongly encouraged to include "UniProtID" column between "Gene Name" and "Spectral Counts" for consistency with the naming scheme in the remaining tabs.

Figure 5: (A) The authors should insert a sentence describing the coloring scheme for the ADPR moiety.

Discussion: In the first paragraph of discussion, the authors speculate there may be an unidentified effector gene that may be proximal to mavL gene. Can the authors state at least one possible candidate based on the genome information?

In the sixth paragraph of discussion, the description may cause some confusion in the readers: did the author intend to say that two different levels of substrate specificity of MavL? The first would be at the residue level where arginine residue is preferred by MavL unlike other similar effectors; the second would be at the substrate protein level where proteins MARYlated at an arginine residue - in this case the authors apparently mentioned "ubiquitin", "Gdp2h" and "Rab5". If this understanding by the reviewer is correct, the authors are strongly encouraged to revise relevant sentences for better flow.

In one sentence in the sixth paragraph of discussion, the authors said "MARYlated Gdp2h" and "ADO-ribosylated Rab5". Are these two modifications the same? If so, please make the terminology consistent. Otherwise, please sort things out.

COMMENTS ON THE ORGANIZATION OF THE REVISED MANUSCRIPT:

Abstract: It would be better if the authors state specifics on "substrate recognition", "catalytic mechanism", and "arginine specificity". For instance, what was the key finding in the arginine specificity.

Results: The authors are strongly advised to align subsections to figures by one-to-one matching. For instance, Figure 1 spans two subsections ("MavL reacts with Ub-derived activity based probes" and "MavL removes ADPR from ADPR-Ub generated by SidE effector family"). It is suggested that these two subsections are merged into one subsection with the title "MavL removes ADPR from ADPR-Ub generated by SidE effector family". Also the two paragraphs of the first subsection are better to be merged into one paragraph.

The subsection titled "DUF4804 as a class of Arg-specific..." is advised to change as "DUF4804 as an eukaryotic MavL orthologous class of Arg-specific...". Is the intention of the authors to form a new class harboring MavL, Larg1, CG2909 and CG3568 as its members?

The subsection title "Structural features unique in Arg..." is better moved one paragraph down so that the previous subsection is aligned with Figure 5 and this subsection with Figure 6.

Discussion: The paragraph describing Figure S6 may be better suited after the third paragraph of the results section. It seems a little bit awkward for a paragraph describing results to appear in the discussion section.

Grammatical errors: The reviewer found some grammatical errors such as incorrect tense and fragmentation of a sentence throughout the revised manuscript. The authors are strongly encouraged to go through the revised manuscript to correct such errors.

Reviewer #2 (Remarks to the Author):

The authors have adequately addressed my concerns.

Reviewer #3 (Remarks to the Author):

The authors have addressed the issues raised by this reviewer satisfactorily overall. However, some issues remain. The reviewer finds that the revised manuscript needs grammatical and organizational touches.

Below are the remaining concerns based on the rebuttal letter and the revised manuscript. Detailed recommendations for manuscript organization are also noted at the end of the comments.

COMMENTS ON THE REMAINING CONCERNS:

Table S1: In the first tab of the EXCEL file provided, the authors are strongly encouraged to include "UniProtID" column between "Gene Name" and "Spectral Counts" for consistency with the naming scheme in the remaining tabs.

Thanks for the comment. We added the UniProtID to the table.

Figure 5: (A) The authors should insert a sentence describing the coloring scheme for the ADPR moiety.

Thanks for the comment. We added a sentence in the figure legend describing the coloring scheme for ADPR.

Discussion: In the first paragraph of discussion, the authors speculate there may be an unidentified effector gene may be proximal to mavL gene. Can the authors state at least one possible candidate based on the genome information?

Thanks for the suggestion. Currently, we don't have a clear idea what could be the possible effector. Therefore, we are reluctant to make any speculation, as it may mislead the

readers. Respectfully, we would like to keep the current discussion unchanged pertaining to this particular point.

In the sixth paragraph of discussion, the description may cause some confusion in the readers: did the author intend to say that two different levels of substrate specificity of MavL? The first would be at the residue level where arginine residue is preferred by MavL unlike other similar effectors; the second would be at the substrate protein level where proteins MARYlated at an arginine residue - in this case the authors apparently mentioned "ubiquitin", "Gdp2h" and "Rab5". If this understanding by the reviewer is correct, the authors are strongly encouraged to revise relevant sentences for better flow.

Thanks for the comments. We modified this paragraph to clear some confusions.

In one sentence in the sixth paragraph of discussion, the authors said "MARYlated Gdp2h" and "ADO-ribosylated Rab5". Are these two modifications the same? If so, please make the terminology consistent. Otherwise, please sort things out.

Thanks for the comment. We made necessary changes to keep the terminology consistent.

COMMENTS ON THE ORGANIZATION OF THE REVISED MANUSCRIPT:

Abstract: It would be better if the authors state specifics on "substrate recognition", "catalytic mechanism", and "arginine specificity". For instance, what was the key finding in the arginine specificity.

Thanks for the comment. We made adjustments along the suggested lines to the abstract to make it more specific.

Results: The authors are strongly advised to align subsections to figures by one-to-one matching. For instance, Figure 1 spans two subsections ("MavL reacts with Ub-derived activity based probes" and "MavL removes ADPR from ADPR-Ub generated by SidE effector family"). It is suggested that these two subsections are merged into one subsection with the title "MavL removes ADPR from ADPR-Ub generated by SidE effector family". Also the two paragraphs of the first subsection are better to be merged into one paragraph.

We merged these two subsections together, with the suggested subtitle.

The subsection titled "DUF4804 as a class of Arg-specific..." is advised to change as "DUF4804 as an eukaryotic MavL orthologous class of Arg-specific...". Is the intention of the authors to form a new class harboring MavL, Larg1, CG2909 and CG3568 as its members?

Thanks for the comment. Our idea is that all these four enzymes belong to the DUF4804 class. Since this class contains both prokaryotic and eukaryotic enzymes, we wish to keep it as it is.

The subsection title "Structural features unique in Arg.." is better moved one paragraph down so that the previous subsection is aligned with Figure 5 and this subsection with Figure 6.

Thanks for the comment. We have made this change.

Discussion: The paragraph describing Figure S6 may be better suited after the third paragraph of the results section. It seems a little bit awkward for a paragraph describing results to appear in the discussion section.

Thanks for the comment. We moved this section back to the results section and adjusted the numbering of figures accordingly.

Grammatical errors: The reviewer found some grammatical errors such as incorrect tense and fragmentation of a sentence throughout the revised manuscript. The authors are strongly encouraged to go through the revised manuscript to correct such errors.

Thank you. We have gone through the manuscript carefully and fixed these errors.